health and disease and epidemiology/ computational biology

contact network, disease ecology, *E. coli*, network epidemiology, real-time location, social network

**Author for correspondence:**
Cristina Lanzas
e-mail: clanzas@ncsu.edu

# Combining epidemiological and ecological methods to quantify social effects on *Escherichia coli* transmission

Trevor S. Farthing[1], Daniel E. Dawson[1], Mike W. Sanderson[2], Hannah Seger[2] and Cristina Lanzas[1]

[1]Department of Population Health and Pathobiology, College of Veterinary Medicine, North Carolina State University, Raleigh, NC 27606, USA
[2]Department of Diagnostic Medicine and Pathobiology, College of Veterinary Medicine, Kansas State University, Manhattan, KS 66506, USA

(iD) TSF, 0000-0003-1169-9913; CL, 0000-0002-4039-0336

Enteric microparasites like *Escherichia coli* use multiple transmission pathways to propagate within and between host populations. Characterizing the relative transmission risk attributable to host social relationships and direct physical contact between individuals is paramount for understanding how microparasites like *E. coli* spread within affected communities and estimating colonization rates. To measure these effects, we carried out commensal *E. coli* transmission experiments in two cattle (*Bos taurus*) herds, wherein all individuals were equipped with real-time location tracking devices. Following transmission experiments in this model system, we derived temporally dynamic social and contact networks from location data. Estimated social affiliations and dyadic contact frequencies during transmission experiments informed pairwise accelerated failure time models that we used to quantify effects of these sociobehavioural variables on weekly *E. coli* colonization risk in these populations. We found that sociobehavioural variables alone were ultimately poor predictors of *E. coli* colonization in feedlot cattle, but can have significant effects on colonization hazard rates ($p \leq 0.05$). We show, however, that observed effects were not consistent between similar populations. This work demonstrates that transmission experiments can be combined with real-time location data collection and processing procedures to create an effective framework for quantifying sociobehavioural effects on microparasite transmission.

## 1. Introduction

Contact- and social-network epidemiology have given us the ability to study drivers of pathogen transmission in unprecedented detail

[1,2]. By constructing networks describing observed interactions or relationships between susceptible and infectious individuals, and annotating demographic, social, weather or season data to network components, researchers can readily examine how these potential drivers affect pathogen transmission [3–8]. Contact networks and social networks are both valuable models for studying pathogen transmission, but they are distinctly different entities [5,9]. Edges in contact networks represent observed coexistence (i.e. individuals exist in the same space at the same time), colocation (i.e. individuals exist in the same space but at different times) or physical interactions between nodes over a defined time period [9]. Edges in social networks, however, denote the existence of specific relationships between nodes (e.g. shared home range, shared parentage, rivalry, friendship, etc.).

For enteric microparasites like *Salmonella* spp. and *Escherichia coli*, which have multiple transmission pathways and reservoirs outside of host organisms [10–13], the relative transmission risk attributable to direct and indirect (i.e. environmental) contacts with infectious individuals or contaminated environments are not wholly understood [7,10,13]. Understanding and accounting for drivers of coexistence, colocation and resource-sharing heterogeneity are paramount to quantifying the contribution of different transmission pathways for estimating enteric pathogen transmission risk to susceptible individuals [14,15]. In recent years, researchers have attempted to infer effects of heterogeneous contact and social networks on enteric pathogen transmission through studies linking observed commensal strain similarity to network edges [16–20]. These studies collectively suggest that direct contact mediated by social interactions are important drivers of enteric microparasite transmission for most species, but often consider the genetic similarity of bacterial commensal isolates to be a proxy for transmission between sampled individuals. However, because animals are often colonized with commensal enteric organisms shortly after birth [21–23], contemporaneous social interactions studied in these cases may be less relevant to the pattern of strain sharing than long-term stable social interactions. Furthermore, these studies often quantify social or contact network influence on transmission, but not effects of both networks simultaneously.

In this paper, we aim to re-evaluate sociobehavioural drivers of enteric microparasite transmission by applying different methods. The use of transmission experiments is an alternative approach for studying enteric pathogen transmission that does not share the limitations of studies examining the spread of endemic microparasites, as discussed above [24]. In transmission experiments, researchers challenge select animals with a unique and traceable infectious agent and release them back into susceptible study populations [25–27]. Study populations are closely surveilled to capture infection or disease incidence over a defined time period. This allows for contemporaneous observation of pathogen transmission between individuals. Because real-time location tracking methods can be used to generate inter-animal contact networks [28–30] and comparisons of observed contact rates can be used to predict sociality within tracked populations [31,32], combining transmission experiments with real-time location data collection provides a powerful framework for gaining inference into sociobehavioural drivers on microparasite transmission.

We carried out an experiment to investigate sociobehavioural effects on transmission for a common enteric organism (*E. coli*) in cattle (*Bos taurus*). Cattle frequently display social behaviours like allogrooming, headbutting, preferential clustering and avoidance [33–35], and commonly host a wide array of enteric pathogens [36–38], making them ideal model populations for studying drivers of enteric pathogen transmission. We challenged five individuals in study populations with a single traceable commensal *E. coli* strain and subsequently allowed transmission to occur naturally in the field. By combining continuous real-time location tracking and longitudinal sampling procedures to frequently evaluate strain prevalence within the herd, we were able to link colonization incidence to temporally dynamic contact and social networks derived from observed animal locations. Because we were able to observe the spread of *E. coli* through our study population, we could infer transmission directionality from known colonized individuals to susceptible ones at every sampling interval. We use these data to quantify how proximity-based contact rates and social relationships contributed to the risk of *E. coli* colonization.

# 2. Methods

## 2.1. Study population and data collection

### 2.1.1. Study population

Two transmission experiments were carried out over two distinct study periods, between 22 May to 10 July 2017 and 21 May to 30 July 2018. We performed experiments over multiple years in an effort

to control for intra-year variation in *E. coli* colonization outcomes due to weather, personalities of the research animals, etc. Transmission experiments were carried out over summer months to maximize the likelihood of observing at least one successful transmission event, as faecal shedding of *E. coli* by cattle, and *E. coli* prevalence in cattle populations are known to be greater during warmer months [39–41]. During each of these periods, 70 approximately 15-month-old beef cattle were introduced to and kept in a single $30.5 \times 38 \, m^2$ outdoor pen at a commercial cattle feedlot research centre in Manhattan, KS. All individuals were castrated males that were unfamiliar with the enclosure prior to entering the study. None of the 70 individuals in the 2017 study was retained for 2018 experiments. The number of individuals included in each transmission experiment (i.e. $n = 70$) was intended to mimic stocking rates observed in US concentrated animal feeding operations, ensuring that observed results reflect real-world colonization rates in these agricultural systems. All animal care, handling and monitoring procedures were approved by the Kansas State University Institutional Animal Care and Use Committee.

### 2.1.2. Longitudinal *Escherichia coli* prevalence data

At the beginning of each study period, five steers were randomly selected using the random number generator in Microsoft Excel 2016 [42] from the pen of 70 for inoculation. To maximize the probability of successfully establishing shedding in inoculated calves, each individual was orally inoculated daily for 5 consecutive days with $10^9$ colony forming units (CFU) of a single *E. coli* strain made resistant to nalidixic acid and rifampicin. The rest of the animals ($n = 65$) were screened for the inoculated *E. coli* with the dual resistance prior to the experiment. Only animals that tested negative were used in the study. We chose to inoculate five individuals because preliminary data suggested that this number would successfully trigger an *E. coli* outbreak in the remaining, non-inoculated population. Faecal samples from all steers were collected weekly for the duration of each study period. Samples were spiral plated on MacConkey agar supplemented with nalidixic acid ($50 \, \mu g \, ml^{-1}$) and rifampicin ($50 \, \mu g \, ml^{-1}$) to quantify the concentration of *E. coli*. Samples negative (i.e. not quantifiable) by spiral plating were enriched in *E. coli* broth for 6 h at 37°C and plated on MacConkey agar supplemented with nalidixic acid ($50 \, \mu g \, ml^{-1}$) and rifampicin ($50 \, \mu g \, ml^{-1}$) to detect the inoculated strain and establish positive or negative shedding status. Any positive (i.e. CFU > 0) faecal samples obtained from individuals that were not initially challenged with *E. coli* were assumed to be the result of successful field transmission of the inoculated strain.

### 2.1.3. Point-location data

To facilitate continuous location tracking over the course of each study period, all calves were outfitted with radio-transmitting ear tags (Smartbow GmbH, Weibern, Austria) that communicated with receivers around the pen. System software triangulated calves' (x, y) positions during each communication event and logged positional data in a central server. The (x, y) coordinate pairs obtained through this real-time location system were 90% accurate to within ± 0.5 m of individuals' true locations according to company documentation (Smartbow GmbH, Weibern, Austria). To lessen error-induced noise and standardize the temporal resolution of our data at 10 s, we filtered and smoothed the data following the procedure we previously outlined in Dawson *et al.* [30]. In accordance with this procedure, prior to smoothing, points that fell outside of the pen area or suggested that individuals were moving in excess of $10 \, m \, s^{-1}$ speeds were assumed to be erroneous and removed from the dataset. We generated a smoothed dataset to be used in subsequent network production, where (x, y) coordinates represented individuals' average location at each 10 s interval over the course of the study period. Properties of the 2017 and 2018 location data are described in electronic supplementary material, Appendix S1.

## 2.2. Network creation

### 2.2.1. Contact networks

Prior to creating contact networks, we removed point locations observed during the first 3 days (i.e. when radio-frequency identification (RFID) tags initiate transmission at different times, and calves are acclimatizing to their new environment) and the last day (i.e. when tags are removed at different times) of each study period from the dataset to prevent abnormal contact patterns associated with these times from biasing results. The spatial threshold for identifying contact has been shown to be influential in

transmission models using point-based locational data and represents a trade-off between the inclusion of biologically relevant behaviour and non-contact noise that can obscure actual contact patterns [30]. Here, we chose 0.71 m as our proximity-based contact threshold because it approximates the estimated maximum distance between two calves' tags during shoulder or chest allogrooming events (0.5 m), while accounting for the positional accuracy of our system [32]. Using location data processing procedures described by Dawson *et al.* [30], we accepted a temporal sampling window of 10 s and created 1104 and 1608 hourly aggregated contact networks for our 2017 and 2018 herds, respectively. Edges in these networks were weighted by the sum of contacts observed between each hourly dyad. Hourly contact networks were used to derive social networks, and later aggregated up to the day level for hazard modelling.

## 2.2.2. Social networks

We derived social networks from contact networks using randomized-path-based methods modified from those described by Spiegel *et al.* [31] who used real-time location data to estimate sociality levels in a sleepy lizard (*Tiliqua rugosa*) population. They showed that by randomizing animals' movement paths (i.e. sequential relocation coordinates) along with discrete temporal blocks (e.g. hour, day, week, etc.), researchers can control for environmental influences on inter-animal contact rates. This is important for isolating social effects on microparasite transmission because social interactions are not the only driver of direct or indirect contacts through which pathogens may be transferred between individuals. Resource and environment sharing also drive contact rates [31,43]. Generating dyadic social networks by adapting the Spiegel *et al.* [31] methods to identify social intent within our contact networks allows us to discount environmental drivers of inter-animal contact rates (e.g. congregating at the food trough during feeding time, or huddling together during rainstorms) in our epidemiological analyses.

   In contrast with our contact networks, where edges describe instances of observed node coexistence, edges in our social networks denote a likely social relationship that explains relatively high contact rates between node pairs (e.g. social affinity). To create social networks, we first generated 100 randomized location datasets by shuffling observed 24 h length individual-level movement paths across the entirety of the empirical dataset (i.e. in randomized sets, calves visited the same locations as they did in the empirical one, but not necessarily on the same day). We then created hourly aggregated null contact graphs by applying our contact network creation procedure to each randomized location set, and averaging observed dyadic contact weights in each hour-length graph across the random replicates. These null graphs were representative of the contact distribution we would expect when all contact events occurred solely due to random chance in any given hour [31].

   Due to differences in the frequency of social behaviours associated with increased dyadic contact rates (e.g. allogrooming, headbutting, etc.) during daytime hours relative to night-time hours—when animals would primarily be immobile and resting—we decided to examine social relationships between calves during daytime hours only, when animals were most active. Figure 1 shows the hourly contact distributions for the 2017 and 2018 herds, and denotes where thresholds for 'active' hours exist each year. Given observed trends in data sparsity and contact behaviour each year, we chose to define active-hour sets for 2017 and 2018 as time points between 06.00.00 to 21.59.59 UTM and 06.00.00 to 19.59.59 UTM, respectively, and subset empirical and null contact models accordingly. Active-hour subsets were then aggregated up to the week level to allow for elucidation of weekly social relationships between calves at the same temporal resolution of *E. coli* sample collection.

   For each week-length graph in empirical and null sets, we calculated the maximum number of potential contacts that could possibly exist between dyadic pairs using the equation

$$\text{potential Contacts}_{ijt} = \sum_{t=1}^{T} \text{presence}_{it} * \text{presence}_{jt}, \tag{2.1}$$

where $t \in T$, the sum number of 10 s windows during 'active' hours over the week, $\text{presence}_{it}$ is a binary variable taking the value 0 if individual $i$ was not observed at time $t$, and 1 if they were, and $\text{presence}_{jt}$ is a binary variable taking the value 0 if individual $j$ was not observed at time $t$, and 1 if they were. We subtracted the number of realized dyadic contacts from potential ones to estimate the number of time points when both dyad members were represented in our empirical and randomized point-location sets, but were not in contact with one another. We used a series of binomial exact tests to compare dyadic in-contact and out-of-contact time point counts in weekly empirical graphs to their null model

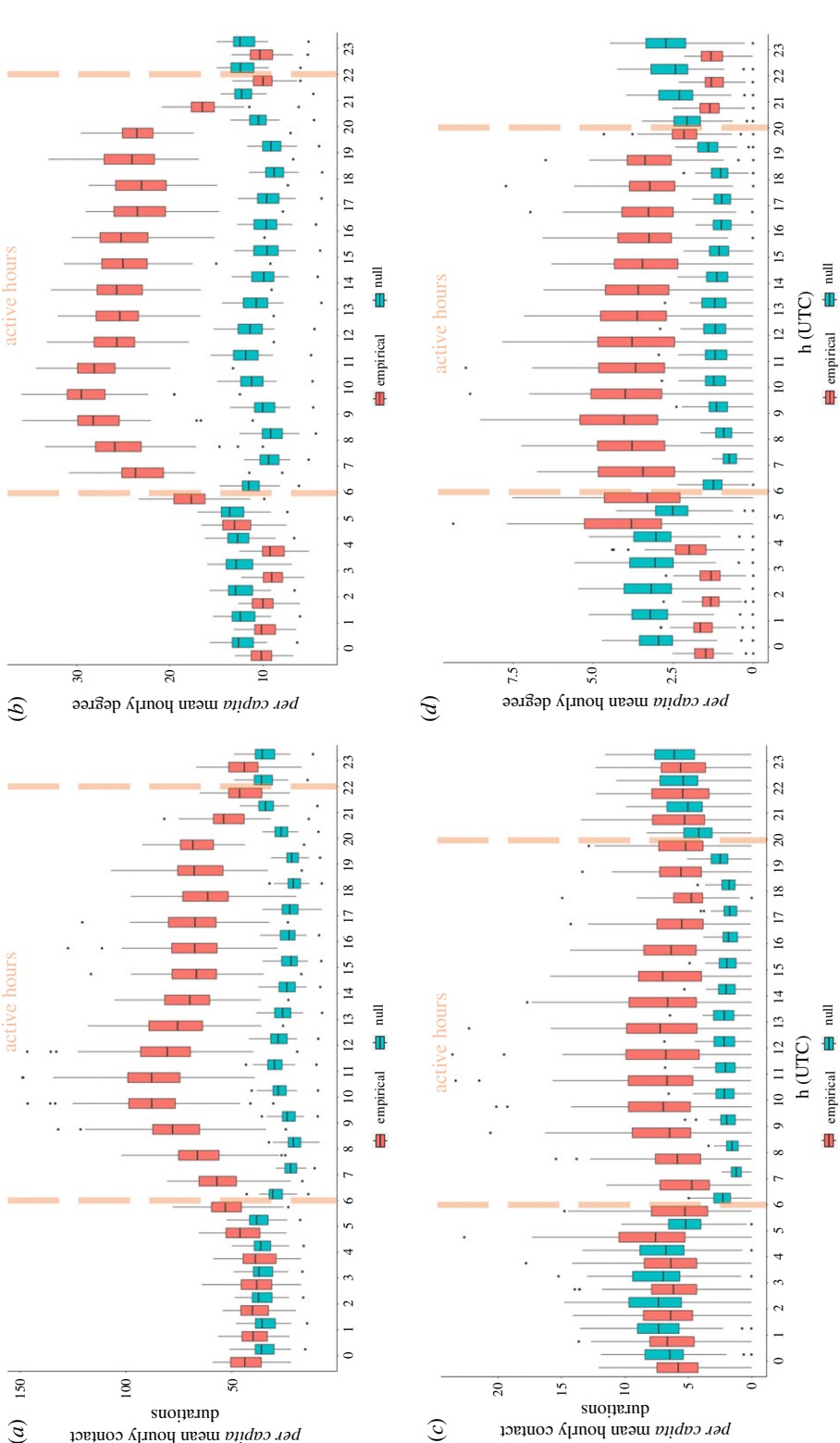

**Figure 1.** Null contact models consistently underpredict *per capita* contact durations and node degree during high-activity hours, but overpredict node degree at other times. (*a*) Comparison of empirical and null model *per capita* observed contact durations in 2017. (*b*) Comparison of empirical and null model *per capita* observed contact durations in 2018. (*d*) Comparison of empirical and null model *per capita* node degree in 2017. (*c*) Comparison of empirical and null model *per capita* node degree in 2018.

counterparts. We chose to use binomial exact tests rather than chi-square goodness of fit tests because 2018 empirical graphs and all null models include instances when expected contact values are very small, and therefore approximations of $p$ given by a chi-square test may not be correct. To control for the type-I error rate associated with running so many tests, we used a Bonferroni-corrected $\alpha$-level to determine 'significant' deviations from null-distribution contact rates. We set $\alpha = 2.07 \times 10^{-05}$ (i.e. $0.05/(70 \times 69)/2$, where the denominator is the number of potential unique dyads in the empirical data). For each empirical week-length 'active' graph, we identified when individuals had more contacts than would be expected at random ($p \leq 2.07 \times 10^{-05}$). When node pairs (i.e. cattle) had more contacts with one another than would be expected at random in a given week, we assigned an edge between them in the weekly social network. Thus, edges in our social networks are indicative of underlying social relationships or behaviours that increased contact frequency between node pairs.

## 2.3. Hazard modelling and analysis

Survival analysis is commonly used to derive infection parameters (e.g. force of infection) from transmission data [25,26,44]. 'Survival analysis' is a broad term describing the suite of statistical methods used for analysing longitudinal time-to-event data [45,46]. Events of interest in survival analysis are commonly referred to as 'failures' and can represent any observable phenomenon (e.g. death, disease onset, etc.) in individuals [46]. Survival data can be censored (i.e. time points when individuals' failure status cannot be known) and truncated (i.e. time points when individuals' were known to have failed at an earlier time, but the failure was not directly observed) [45,46]. Traditional survival methods assume that observed failure events are independent from one another, but this assumption is violated when the event of interest is related to infectious disease transmission [47,48]. A recently developed branch of survival analysis—pairwise survival analysis—specifically intended for modelling infectious diseases makes no such assumption [44,47,49–51]. These methods model failure risk over discrete contact intervals during which a specific susceptible individual is at risk of being infected by a paired infectious individual [47–52]. In short, these methods do not assume that all individuals in a population have the same baseline hazard function (i.e. probability distribution describing the likelihood that individuals surviving until time $t$ will fail before time $t + 1$) with potential scaling factors that allow for individual-level failure time variation. Rather, these methods instead assume that dyads, not individuals, have identical baseline hazard functions and scaling factors that allow for *dyad*-level variation [44,48].

We used the parametric pairwise accelerated failure time (AFT) regression modelling procedure presented by Kenah [48] to estimate the effect of covariates on individual-level weekly *E. coli* colonization hazard. To generate pairwise survival data for our system, we defined failures (i.e. events of interest—here, observed *E. coli* colonization—in survival analysis) as instances when individuals were first observed having a positive faecal sample. Therefore, the discrete contact interval unit for which we evaluate pairwise hazard rate ratios was one week. Individuals were said to be susceptible at interval $\tau$ if they had not presented a positive culture at any previous time. We assumed a latent period of less than 6 days, so individuals observed failing are assumed to have been colonized at some point following the previous sampling date. Because individuals could have been colonized at any point during a given week, we defined shedding individuals at interval $\tau$ as those that had a positive faecal sample in the current week (i.e. interval $\tau$) or in the previous week (i.e. interval $\tau - 1$). We assume these individuals were shedding CFU into the environment between sample collections. Because who-infected-whom could not be known in detail, all susceptible individuals at any given $\tau$ were assumed to have an equal probability of being colonized by any other known shedding individuals at that interval. Susceptible individuals observed failing at interval $\tau$, however, had a 0% probability of colonizing themselves. In the rare event that no sample was collected from specific individuals in a given week, their survival status was truncated from interval $\tau - 1$ to $\tau + 1$. Though we technically monitored individuals for 6 and 10 weeks in 2017 and 2018, respectively, because the pairwise AFT modelling methodology assumes that colonization risk to susceptible individuals only exists when shedding individuals are or will be present to propagate the epidemic [48], we can only use this procedure to model hazard rates up until the final time point when shedding individuals are observed. This occurred in weeks 5 and 10 of 2017 and 2018, respectively. Therefore, for 2017, we were only able to model hazard rates through week 5.

For each shedding individual, susceptible individual pair, $ij$, the pairwise hazard rate parameter, $\lambda_{ij}$, describing the risk of successful transmission of a pathogen from $i$ to $j$ is

$$\ln\lambda_{ij}(\tau,X) = \ln\lambda_0(\tau)\ \beta_k^{\mathrm{T}} X_{ij}. \tag{2.2}$$

**Table 1.** Description of covariates included in hazard models.

| name | type | definition |
|---|---|---|
| contacts | pairwise | average number of daily proximity contacts, obtained from contact networks, between individuals $i$ and $j$ during the time interval $t-1$ to $t$. |
| $degree_i$ | infectiousness | degree of individual $i$ in the week $t$ social network. |
| $degree_j$ | susceptibility | degree of individual $j$ in the week $t$ social network. |
| social | pairwise | binary covariate describing if an edge exists between individuals $i$ and $j$ in the week $t$ social network. |

In this equation, $\lambda_0$ is an unknown baseline dyad-level hazard rate, $\beta_k$ is an unknown coefficient vector of length $X_{ij}$, and $X_{ij}$ is a covariate vector that can contain any combination of infectiousness covariates (i.e. relating solely to individual $i$), susceptibility covariates (i.e. relating solely to individual $j$) and pairwise covariates [44]. Each coefficient in $\beta_k$ can be considered to be the log rate ratio, relative to $\ln\lambda_0$, associated with a one-unit increase of the corresponding covariate in $X_{ij}$.

We used parametric Weibull survival functions to fit the data, where trends in pairwise hazard rates were explained by one to four covariates of interest. A description of each covariate is given in table 1. The contacts covariate was included to test the hypothesis that *E. coli* transmission is driven by physical contacts with shedding individuals, and social was included to determine if the effect of each direct contact on transmission risk varies in accordance with the social relationship between dyad members. We included the $degree_i$ and $degree_j$ covariates to test the respective hypotheses that shedding individuals' probability of infecting others, and susceptible individuals' probability of being colonized correlates with their weekly social network degree. Because weekly hazard rate estimates for susceptible individuals assume they remain uncolonized at interval $\tau$, and we cannot assess dyadic social covariate values for interval $\tau = 0$ (i.e. the interval from $t=-1$ to $t=0$, which precedes the beginning of the study period), without conflating structural and sampling zeroes in our data, we could only estimate weekly hazard rates for individuals remaining uncolonized until $\tau \geq 2$.

We used an Akaike information criterion (AIC)-based model selection approach to evaluate the assembled hypotheses. First, we fit the model including all terms:

$$\ln\lambda_{ij}(\tau,X) = \ln\lambda_0(\tau)\,(\beta_1\text{contacts} + \beta_2\text{social} + \beta_3\text{degree}_i + \beta_4\text{degree}_j + \beta_5\text{contacts} \times \text{social}). \quad (2.3)$$

Then, we fit all potential nested variants ($n=40$). A preliminary analysis of Pearson's correlation coefficients between covariate pairs indicated that no pair had a correlation value greater than 0.5, our correlation threshold for minimizing collinearity issues in the fitted models. We used the Nelder–Mead algorithm [53] to estimate maximum likelihood for model parameters. For all Weibull distributions, the probability density function was

$$f(x) = \lambda\theta(\lambda\tau)^{\theta+1}e^{-(\lambda\tau)^\theta}, \quad (2.4)$$

where the shape parameter ($\theta$) was assumed to be the same for all $ij$ pairs, but the rate parameter ($\lambda$) was allowed to vary [48]. Due to aforementioned differences in point-location data quality between 2017 and 2018 study periods, observed contact and social metrics are not directly comparable. Thus, we modelled hazard rates in each year separately. For each year, we report models with AIC weights greater than or equal to 5% and describe Wald 95% confidence intervals and $p$-value estimates for pairwise hazard ratios determined from best-fitting models (i.e. model with the greatest weight each year).

We define candidate model sets for 2017 and 2018 data as $M_1$ and $M_2$, respectively. To evaluate the overall fit of models in our two yearly candidate sets, $m \in M$, we used model averaging [54] to compile a most-likely weekly cumulative survival probability, $S(\tau)_{jM}$, for each susceptible individual observed during the study period. These survival probabilities represent the likelihood that individual $j$ was not colonized with *E. coli* on or before sampling interval $\tau$. Cumulative survival probabilities were

estimated using the equation

$$S(\tau)_{jM} = \sum_{m=1}^{M} w_m (1 - F(\tau)_{jm})$$

$$F(\tau)_{jm} = 1 - e^{-\left(\left(\sum_{i=1}^{I_\tau} \lambda_{ij}\right)\tau\right)^{\theta_m}},$$

(2.5)

where $F(\tau)_{jm}$ is the cumulative failure time distribution function for susceptible individual $j$ at interval $\tau$, as predicted by a given model ($m$) with a Weibull distribution and shape parameter $\theta_m$, $\sum_{i=1}^{I_\tau} \lambda_{ij}$ is the sum of pairwise hazard rates acting on individual $j$ at interval $\tau$ and $w_m$ is the model-specific AIC weight. These individual-level survival probabilities were used to weight random draws of binary *E. coli* colonization status for each *E. coli* sampling interval in 100 000 bootstrap replicates. All random draws in bootstrap replicates were independent of another. To assess the predictive ability of our models, we calculated the mean population-level cumulative survival probability and the 95% bootstrap percentile confidence interval at each *E. coli* sampling interval from bootstrap estimates, and plotted these together with the empirical cumulative survival rates.

## 2.4. Software

All data processing and analyses were carried out in RStudio (v. 1.4.1103, RStudio Team, Boston, MA [55]) running R (v. 4.0.3, R Foundation for Statistical Computing, Vienna, Austria [56]). We used the *contact* package (v. 1.2.6), introduced in Farthing *et al.* [32] to clean point-location data and create contact and social networks. We used the *transtat* package (v. 0.3.4, available at github.com/ekenah/transtat [48]) for all pairwise hazard modelling and analysis.

# 3. Results

## 3.1. Yearly metric comparisons

Weekly contact and social network descriptions and connectivity metrics are described in tables 2 and 3, respectively. As shown in table 2, mean daily observed proximity-based contact rates for individuals connected in weekly contact networks were approximately equal to 10 times greater in 2017 than in 2018. Average weekly node degree in social networks, however, tended to be much greater in 2018 (table 3). In 2018, *ij* node pairs were approximately equal to four times more likely to be connected by edges in weekly social networks than in 2017, on average. These inter-year differences were probably largely due to aforementioned disparities in location data quality between the years, which reduced the probability that tracked individuals in 2018 were observed concurrently at each timestep relative to the 2017 study population. When individuals were unable to be observed concurrently on a given timestep, no proximity-based contact events could be identified for the pair. Subsequently, because 2018 dyads had relatively low edge weights in weekly contact networks, relatively fewer contact events were required to characterize significant ($p \leq 2.07e^{-05}$) social affiliations. This resulted in weekly social networks that were much denser than those observed in 2017.

## 3.2. Pairwise accelerated failure time models

Weekly colonization rates were similar in both study periods (figure 2a). In 2017, four individuals were found to be colonized after the first contact interval ($\tau = 1$). Thus, weekly hazard rate estimates are based only on potential *E. coli* transmission to the remaining 61 susceptible calves. In 2018, 32 individuals were found to be colonized at the first sampling period.

We report candidate AFT models for explaining variation in *E. coli* colonization with AIC weights ($w_m$) ≥ 5% in tables 4 and 5. Tables of AIC metrics for all candidate models in 2017 and 2018 can be found in electronic supplementary material, Appendix S2. There were nine models from the 2017 set with AIC weights greater than or equal to 5%, but no model containing our covariates of interest outperformed the null model (i.e. $\lambda_{ij}(\tau) = \lambda_0(\tau)$). A ratio of AIC weights indicates that there is 1.66 times more empirical support for the null model than the second-best-fitting model for explaining *E. coli* colonization rates in 2017, but the predictive ability of the null model is relatively poor after the first week of the study period (figure 2b). Regarding 2018 data, there were five candidate models with

**Table 2.** Summary metrics for weekly inter-calf contact networks in 2017 and 2018. Edges in contact networks indicate that animals' real-time location points were observed within 0.71 m of one another at least one time during the week.

| year | week | nodes (total) | nodes (shedding) | nodes (susceptible)[a] | edges | density | edge weight (mean daily contacts)[b] | median node degree[c] | median node betweenness[c] |
|---|---|---|---|---|---|---|---|---|---|
| 2017 | 1 | 70 | 9 | 64[d] | 2415 | 1 | 24 | 69 | 0 |
| | 2 | 70 | 35 | 61 | 2415 | 1 | 22 | 69 | 0 |
| | 3 | 70 | 37 | 32 | 2415 | 1 | 22 | 69 | 0 |
| | 4 | 70 | 47 | 29 | 2415 | 1 | 23 | 69 | 0 |
| | 5 | 70 | 47 | 13 | 2415 | 1 | 21 | 69 | 0 |
| | 6 | 70 | 0 | 13 | 2415 | 1 | 19 | 69 | 0 |
| 2018 | 1 | 70 | 37 | 65 | 2249 | 0.93 | 2.6 | 66 | 1.5 |
| | 2 | 70 | 42 | 33 | 2309 | 0.96 | 2.2 | 67 | 0.63 |
| | 3 | 70 | 28 | 26 | 2309 | 0.96 | 3.1 | 68 | 0.64 |
| | 4 | 70 | 21 | 20 | 2244 | 0.93 | 2.6 | 67 | 0.6 |
| | 5 | 70 | 20 | 19 | 2248 | 0.93 | 2.8 | 67 | 0.48 |
| | 6 | 70 | 42 | 16 | 2166 | 0.90 | 2.6 | 65 | 0.77 |
| | 7 | 70 | 45 | 11 | 2125 | 0.88 | 2.4 | 65 | 0.71 |
| | 8 | 70 | 35 | 7 | 2073 | 0.86 | 1.9 | 64 | 1.2 |
| | 9 | 70 | 14 | 6 | 2076 | 0.86 | 2.3 | 64 | 1.1 |
| | 10 | 70 | 5 | 6 | 2092 | 0.87 | 2 | 64 | 0.86 |

[a]The number of susceptible individuals at risk of being colonized going into the sampling period. 'At risk' individuals can be observed shedding E. coli during the weekly sampling period if they became colonized during the week, and therefore, lists of shedding and susceptible nodes are not mutually exclusive.

[b]Edge-level variable indicating the mean number of times each dyad was observed in contact (i.e. within 0.71 m of one another) during the week.

[c]These are node-level variables.

[d]There were a total of 65 susceptible individuals initially at risk in 2017, but the sample for one individual was missing in week 1 so that individual was censored until week 2.

**Table 3.** Summary metrics for weekly inter-calf social networks in 2017 and 2018. Edges in social networks indicate that animals shared more average daily contacts (i.e. instances when their real-time location points were within 0.71 m of one another) than would be expected at random.

| year | week | nodes (total) | nodes (shedding) | nodes (susceptible)[a] | edges | density | median node degree[b] | median node betweenness[b] |
|---|---|---|---|---|---|---|---|---|
| 2017 | 1 | 70 | 9 | 64[c] | 510 | 0.21 | 12 | 12 |
| | 2 | 70 | 35 | 61 | 251 | 0.1 | 5 | 3.9 |
| | 3 | 70 | 37 | 32 | 218 | 0.09 | 4 | 7 |
| | 4 | 70 | 47 | 29 | 390 | 0.16 | 8 | 2.9 |
| | 5 | 70 | 47 | 13 | 326 | 0.13 | 8 | 6.8 |
| | 6 | 70 | 0 | 13 | 411 | 0.17 | 11 | 11 |
| 2018 | 1 | 70 | 37 | 65 | 941 | 0.39 | 29 | 13 |
| | 2 | 70 | 42 | 33 | 1012 | 0.42 | 29 | 15 |
| | 3 | 70 | 28 | 26 | 1138 | 0.47 | 32 | 15 |
| | 4 | 70 | 21 | 20 | 1069 | 0.44 | 32 | 17 |
| | 5 | 70 | 20 | 19 | 1279 | 0.53 | 39 | 15 |
| | 6 | 70 | 42 | 16 | 1284 | 0.53 | 39 | 14 |
| | 7 | 70 | 45 | 11 | 1121 | 0.46 | 35 | 15 |
| | 8 | 70 | 35 | 7 | 1176 | 0.49 | 36 | 12 |
| | 9 | 70 | 14 | 6 | 1168 | 0.48 | 36.5 | 14 |
| | 10 | 70 | 5 | 6 | 1234 | 0.51 | 39.5 | 13 |

[a]The number of susceptible individuals at risk of being colonized going into the sampling period. 'At risk' individuals can be observed shedding *E. coli* during the weekly sampling period if they became colonized during the week, and therefore, lists of shedding and susceptible nodes are not mutually exclusive.
[b]These are node-level variables.
[c]There were a total of 65 susceptible individuals initially at risk in 2017, but the sample for one individual was missing in week 1 so that individual was censored until week 2.

$w_m \geq 5\%$ (table 4). The 2018 candidate with the greatest $w_m$ is

$$\ln\lambda_{ij}(\tau,X) = \ln\lambda_0(\tau)\,(\beta_1\mathrm{degree}_i + \beta_2\mathrm{degree}_j). \tag{3.1}$$

The $w_m$ for this model is 0.29, and evidence ratios indicate that there is 64.63 and 1.50 times more empirical support for this model than for the null or second-best-fitting 2018 model, respectively. This model suggests that susceptible individuals with greater social degrees ($\mathrm{degree}_j$) were at greater risk of being colonized, while shedding individuals with greater social degrees ($\mathrm{degree}_i$) were less likely to transmit *E. coli* to others (table 3). Despite significant ($p \leq 0.05$) effects outlined by the best-fitting 2018 model, bootstrap estimates of the cumulative survival probability at each sampling interval in 2018 indicate that the models we fit were unable to accurately predict *E. coli* colonization risk during the final two weeks of the study period (figure 2c). Because all elevated predicted values are associated with *E. coli* sampling intervals during which relatively few shedding individuals were observed (tables 1 and 2), it is possible that these inaccuracies arise due to consistent underprediction of the baseline dyad-level hazard rate, $\lambda_0$, by large-weight models in the 2018 candidate set.

## 4. Discussion

This study evaluates sociobehavioural predictors of *E. coli* transmission within the time scale of an outbreak (less than year). We used pairwise AFT modelling on an integrated dataset generated from a combination of *E. coli* field transmission studies and continuous animal point-location tracking to

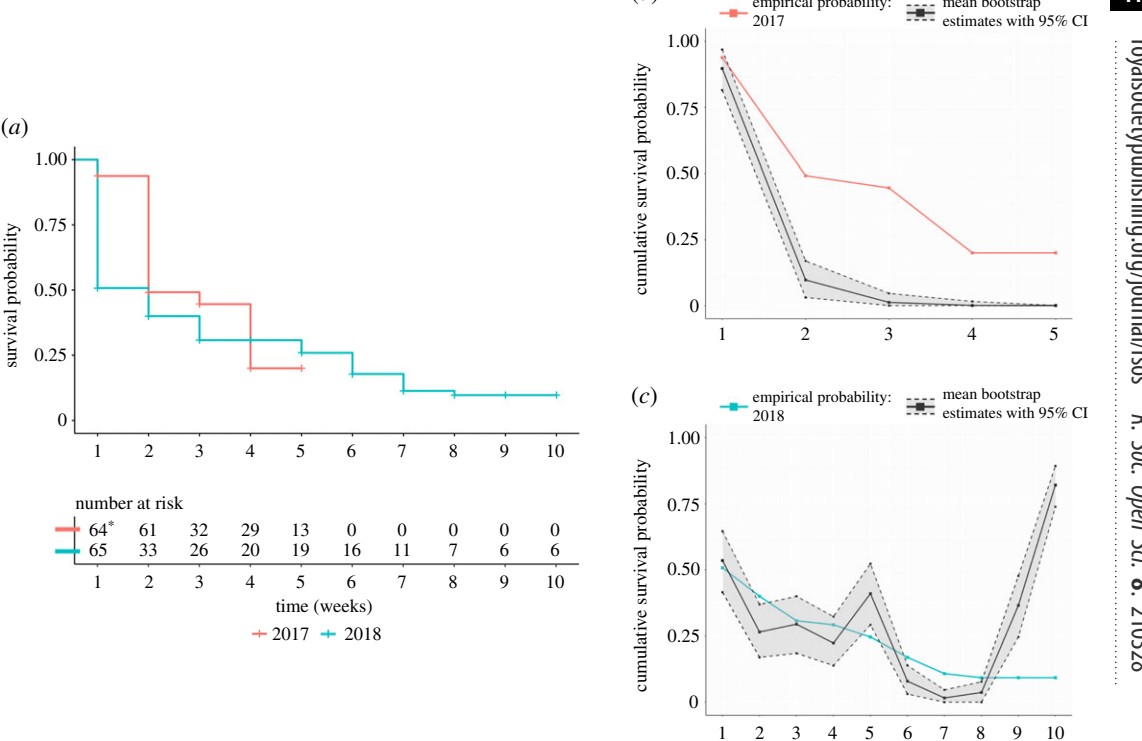

**Figure 2.** Empirical survival curves show that *E. coli* transmission in 2017 and 2018 occurred at similar rates. (*a*) Proportion of 2017 and 2018 study populations observed remaining uncolonized by *E. coli* at weekly timesteps. 'At risk' numbers indicate the number of individuals that had not been colonized any time prior to the given week (e.g. if 65 individuals were at risk at time 1, but only 33 were at risk at time 2, then 32 individuals were found to be colonized at time 1). Asterisk represents that there were a total of 65 susceptible individuals initially at risk in 2017, but the sample for one individual was missing in week 1. That individual was censored until week 2. (*b*) Comparison of empirical and model-derived estimates of weekly population-level cumulative survival probabilities in 2017. Mean estimates derived from 100 000 bootstrap samples are shown together with 95% bootstrap percentile confidence intervals. (*c*) Comparison of empirical and model-derived estimates of weekly population-level cumulative survival probabilities in 2018. Mean estimates derived from 100 000 bootstrap samples are shown together with 95% bootstrap percentile confidence intervals.

quantify relative effects attributable to social covariates on *E. coli* colonization risk among cattle. Our results suggest that the number of close-proximity contacts and social relationships with shedding individuals are poor predictors for *E. coli* colonization in susceptible feedlot cattle overall, but can influence colonization rates. Furthermore, our high-weight AFT models for predicting colonization rates during the 2018 study period support the hypothesis that the number of dyadic social partners with whom an individual associates can modulate their *E. coli* colonization and transmission risk.

Candidate models that incorporated effects of contact rates or social affiliations with shedding individuals did fit our 2018 data better than the null model, suggesting that social structure and frequent physical contact between animals may have driven *E. coli* transmission in this population. This finding contributes to the growing body of evidence that enteric pathogen incidence within affected populations may be the result of physical interactions with socially related infectious individuals [17,20,57]. Given that the null model was the best-fitting 2017 candidate, however, effects of social relationships and direct contact rates on *E. coli* transmission may be highly variable even in similar populations.

We suspect that the ability of our models to predict weekly survival rates for susceptible individuals (figure 2*b,c*) could be improved with the addition of covariates that describe environmental conditions (e.g. rainfall, temperature, soil CFU load, etc.) during each contact interval ($\tau$). The ability of *E. coli* to survive in soil for months without hosts [58,59] increases the likelihood that environment-mediated transmission contributes to new colonization cases [13], and environmental covariate effects on *E. coli* carriage and soil concentrations have been previously documented [60–63].

Unfortunately, pairwise AFT models parametrized like those we used here are ill-suited for incorporating covariates without individual- or dyad-level variation, as the model equations imply

**Table 4.** Covariates in candidate AFT models with AIC weights greater than or equal to 5%.

| model terms[a] | AIC | $\Delta$AIC | $w_m$ | $w_m / w_{null}$ |
|---|---|---|---|---|
| **2017** | | | | |
| $\ln\lambda_0(\tau)$ | 228.81 | 0 | 0.22 | 1.00 |
| $\beta_1$contacts | 229.82 | 1.01 | 0.13 | 0.60 |
| $\beta_1$degree$_j$ | 230.39 | 1.58 | 0.11 | 0.45 |
| $\beta_1$degree$_i$ | 230.68 | 1.87 | 0.09 | 0.39 |
| $\beta_1$social | 230.68 | 1.87 | 0.09 | 0.39 |
| $\beta_1$contacts $+ \beta_2$social $+ \beta_3$contacts $\times$ social | 231.45 | 2.64 | 0.06 | 0.27 |
| $\beta_1$contacts $+ \beta_2$degree$_i$ | 231.79 | 2.98 | 0.05 | 0.23 |
| $\beta_1$contacts $+ \beta_2$social | 231.80 | 2.99 | 0.05 | 0.22 |
| $\beta_1$contacts $+ \beta_2$degree$_j$ | 231.81 | 3.00 | 0.05 | 0.22 |
| **2018** | | | | |
| $\beta_1$degree$_i + \beta_2$degree$_j$ | 249.48 | 0 | 0.29 | 64.63 |
| $\beta_1$contacts $+ \beta_2$degree$_i + \beta_3$degree$_j$ | 250.29 | 0.81 | 0.19 | 43.12 |
| $\beta_1$degree$_i$ | 250.36 | 0.88 | 0.19 | 41.71 |
| $\beta_1$social $+ \beta_2$degree$_i + \beta_3$degree$_j$ | 251.64 | 2.16 | 0.10 | 21.96 |
| $\beta_1$social $+ \beta_2$degree$_i$ | 251.92 | 2.44 | 0.09 | 19.08 |

[a]All AFT models were reported in the form: $\ln\lambda_{ij}(\tau,X) = \ln\lambda_0(\tau)\ \beta_k^{\mathsf{T}} X_{ij}$. For simplicity, aside from the null model, we only list terms included in the $\beta_k^{\mathsf{T}} X_{ij}$ vector here.

**Table 5.** Coefficients and rate ratios associated with 1-unit increases in covariate values given by best-fitting pairwise AFT models. Wald 95% confidence intervals are given in parentheses. All AFT models were reported in the form: $\ln\lambda_{ij}(\tau,X) = \ln\lambda_0(\tau)\ \beta_k^{\mathsf{T}} X_{ij}$.

| covariate | estimated coefficient ($\boldsymbol{\beta}$) | rate ratio | $p$ |
|---|---|---|---|
| **2017** | | | |
| $\ln\theta$[a] | 0.396 (0.069, 0.723) | — | <0.001 |
| $\ln\lambda_0$ | −3.697 (−4.423, −2.972) | — | 0.018 |
| **2018** | | | |
| $\ln\theta$[a] | 0.327 (0.148, 0.506) | — | <0.001 |
| $\ln\lambda_0$ | −2.740 (−3.270, −2.211) | — | <0.001 |
| degree$_i$ | −0.079 (−0.115, −0.042) | 0.924 (0.892, 0.959) | <0.001 |
| degree$_j$ | 0.023 (0.001, 0.044) | 1.023 (1.001, 1.045) | 0.039 |

[a]This is not a model covariate, but rather the estimated natural log of the Weibull distribution's shape parameter.

that such covariates would be weighted by the instantaneous number of infectious-susceptible pairs at each contact interval [48]. Sharker & Kenah [44] present methods for pairwise AFT models of systems with direct transmission routes (i.e. known infectious individuals are able to infect susceptible ones) and unobserved 'external' infectors, which do allow for the inclusion of population-stable variables to modulate environment-related hazard. However, their methods assume that effects of direct and external transmission on hazard rates are independent. This assumption is violated in our microparasite–host system because we expect environment-mediated transmission risk to fluctuate as colonized individuals shed *E. coli* CFUs. In addition to the findings we present here, our work highlights the need for further statistical method development in this area.

There were a number of limitations in our study that we must acknowledge. First, calves in our study were always confined to their enclosure. Cattle and various other species are known to display faeces-avoidance behaviours [64,65], but it is possible that confined calves in our study may have been less

able to avoid contaminated faeces. This may have inflated *E. coli* transmission rates to levels unlikely to be seen in free-ranging or wild populations. Second, we carried out transmission experiments only during summer months. Cattle are known to show increased *E. coli* shedding and carriage rates in summer months [39–41,66] and may alter their behaviours (e.g. reduced activity and feed consumption, increased water consumption, etc.) in such a way that may change contact or social network structures in hot weather as well [67,68]. Therefore, the transmission dynamics we observed may differ from those expected in other seasons. Third, we evaluated the transmission of a commensal organism. The commensal behaviour of the generic *E. coli* in cattle facilitated the study of heterogeneities related to exposure and made transmission pathways easier to document because high variation among host individuals in resistance to colonization is less likely to have evolved and become a dominant source of heterogeneity as in pathogens [16,17]. However, for pathogens, illness-induced behavioural effects can disrupt normal inter-animal interaction rates and alter within-population transmission dynamics [69,70]. Thus, social and contact covariate effects may differ for pathogenic microparasite strains.

Finally, we also relied on real-time point-location data to ascertain contact and social information in our study populations. As a result, our study is subject to the inherent inaccuracies and noisiness of the real-time location system's triangulation ability [32]. Procedures used to process the point-location data and define contact and social events have a direct effect on disease modelling outcomes [30]. Observed effects of physical contact and social behaviours on hazard rate estimates are therefore intrinsically linked to our point-location data filtering and contact-identification methods. It should additionally be noted that we lack the ability to identify what specific social relationships were actually represented in our social networks. Edges in our social networks just indicate which node pairs had noticeably elevated contact rates relative to null models. Therefore, we cannot discount the possibility that different relationship types associated with increased physical contact rates (e.g. social affinity, physical aggression, etc.) occurred at varying frequencies across both years and affected colonization risks differently.

Limitations aside, we were able to demonstrate that transmission experiments can be combined with real-time location data collection and processing procedures to create an effective framework for quantifying social forces on microparasite transmission. Additionally, we illustrated how ecological methods for identifying sociality in animal populations can be adapted to characterize specific social relationships between individuals. Our findings ultimately highlight the need for studies seeking to quantify sociobehavioural effects on microparasite transmission dynamics to deploy longitudinal sample collection regimes and collect samples from multiple populations, so as to capture a more-complete range of effect values which may be highly variable over time. Furthermore, our findings suggest that researchers should be wary of assuming homogeneous mixing rates (i.e. all individuals interact with the same probability) and associated effects on transmission when estimating enteric pathogen infection rates in animal populations, even in highly connected populations. This assumption is often made when effective contact rates are unknown, but is only valid when contact patterns are random or near-random, or when infection risk is independent of direct contact rates [71,72]. Here, not only have we shown that animals in highly connected contact networks can have preferential social affiliations with associated differences in contact rates, but also that these variables can affect *E. coli* colonization rates in some cases. It is our hope that future studies can build on our work to identify hallmark predictors of indirect and direct enteric pathogen transmission in varied disease-host systems.

Ethics. All animal care, handling and monitoring procedures were approved by the Kansas State University Institutional Animal Care and Use Committee.

Data accessibility. All data and code used to inform and generate survival models are available in Dryad ([73]—https://doi.org/10.5061/dryad.vt4b8gtsr) and Zenodo ([74]—https://doi.org/10.5281/zenodo.5395180) repositories, respectively.

Authors' contributions. Regarding specific author contributions, C.L. and M.W.S. conceived the ideas presented here and secured funding. M.W.S. and H.S. managed animals and collected *E. coli* samples. T.S.F. and D.E.D. carried out network creation and survival analyses. T.S.F. led the writing of the manuscript, but all authors contributed to the drafts and gave final approval for publication.

Competing interests. We declare we have no competing interests

Funding. This work was supported by US National Institutes of Health (NIH) grant no. R01GM117618 as part of the joint National Science Foundation-NIH-United States Department of Agriculture Ecology and Evolution of Infectious Disease programme.

Acknowledgements. We also want to thank the peer-reviewers who took the time to review our work. Their efforts led to drastic improvements in the quality and clarity of this manuscript.

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
