## [Peer Review File · Royal Society Open Science]

Review History

RSOS-210328.R0 (Original submission)

Review form: Reviewer 1

Is the manuscript scientifically sound in its present form?

No

Are the interpretations and conclusions justified by the results?

No

Is the language acceptable?

Yes

Do you have any ethical concerns with this paper?

No

Have you any concerns about statistical analyses in this paper?

Yes

Recommendation?

Major revision is needed (please make suggestions in comments)

Comments to the Author(s)

This is well written analysis of an ambitious and exciting attempt to quantify the contribution of physical and social network structure on the transmission of *E. coli* between cattle under realistic production conditions. High resolution spatial tracking data is used to characterise dynamic contact and social networks within two experimental groups of cattle. This network data is then used to attempt to tease apart the direct and indirect contributions to the natural transmission of *E. coli* (following experimental inoculation of a small number of seeder animals). The manuscript is supported by an exemplary commitment to open science, with raw data, processed data sets and code made available for review which greatly assisted this process.

However, I have some major concerns about both the formulation of the hazard models that provide the central results and the (lack) of robust model checking which I suspect means that models which have failed to converge have been retained for model selection (and in the case of the 2017 data set selected as the best fitting model).

The pairwise accelerated failure time regression models as implemented in the transtat package are an astute and appropriate framework with which to model this data and address the authors central research question. However, I am concerned about the biological interpretation of some of the covariate terms given how the regression terms are both described in the manuscript (equation 3) and implemented in the associated online code. Equation 3 implies that environmental co-variates (I_j , penSample, precip, social) contribute an equal (multiplicative) factor to the risk of transmission between each susceptible-infected pair within the herd. As such, are not each of these environmental variables being implicitly weighted by the instantaneous number of infectious-susceptible pairs at each time point? My understanding is that the number of infectious individuals at each time is already implicitly accounted for by the baseline hazard (intercept term)? By adding I_j as an separate covariate is this term not effectively introducing a non-linear hazard response?

In relation to this issue I also note that the preprint by Sharker & Kenah (2019) cited by the authors describes a mixed modelling approach to include both "internal transmission" (as represented by the dyadic interactions modelled here) and "external transmission" terms (including environmental covariates). This would appear to be ideal for your use case, but is not what you describe in the manuscript or apparently how your code is functioning.

If my understanding is correct and the intercept and I_j terms are effectively modelling the same effect, one might expect this to lead to identifiability or convergence issues for the MLE procedure. Examining the best fit model for the 2017 data set does indeed suggest a convergence issue as the variance estimate for one of the model parameters is negative. The online source code suggests this may be a bug with the reporting of the transtat package leading to missing confidence intervals. However, I would note that the confidence intervals can be calculated using the slower likelihood ratio method (rather than the Wald approximation) and are infinite, suggesting that in these cases the parameter estimates are singular rather than this being a reporting (or rounding issue). There is further evidence of convergence issues for these models given that different optimisation routines return wildly different parameter estimates (and associated log-likelihoods).

These symptoms, I would suggest, are most likely to mean that the models demonstrating these issues are misspecified or poorly fit the data. The log-likelihood values presented only give a relative measure of fit based on comparison between the alternative models explored and do not provide a quantitative measure of the extent to which the fitted models can explain the original data set. The exceptionally high (weighted mean) hazard rates presented in Figure 4 for the 2017 models seem biologically implausible given the dramatically smaller number of colonisation events in the first two weeks for the 2017 experiment compared to 2018. This again suggests either convergence or fit issues to the 2017 data, raising the possibility that the hazard models simply can't explain the observed data for 2017 given the available covariates. In addition to plotting the hazard rates derived from the models it would be useful to see the predicted survival curves from the fitted models.

In conclusion, this is an exciting and novel data set and application of a cutting edge statistical analysis that could be a terrific contribution to the field if the concerns above can be addressed with further evidence of appropriate model checks.

Minor comments

- 1) I would consider the derived social and contact networks in the groups to be of intrinsic interest themselves and was disappointed not to have seen them compared (or at least visualised!). The procedure to identify social links is a pragmatic approach that makes sense algorithmically. But are the network properties of the inferred social networks consistent with those found in observational studies of cattle behaviour? Also, how do the properties of the social network differ from the contact networks?
- 2) Figure 3, there are no uncertainty intervals plotted on the survival curve
- 3) The difference in the temporal sampling of animals positions in the two years of the study is described in multiple places in the manuscript (and in detail in supplemental information). However, the reason for this difference is not given. Was this an intentional change in the sampling frequency for operational or logistic reasons, or was it the result of losing a greater number of records due to filtering?
- 4) With respect to the difference in sampling frequency between the two years, would be worth considering doing a sensitivity analysis downsampling the 2018 data to show how/if the reduced resolution impacts on the measured contact and social networks
- 5) Line 245, I am a little confused by the implication that the 2017 data set was terminated early due to two weeks in a row with 0 positive fecal samples. Should this not be 0 negative fecal samples, or was *E. coli* eliminated from this herd? Other figures and data suggest not.

Review form: Reviewer 2

Is the manuscript scientifically sound in its present form?

Yes

Are the interpretations and conclusions justified by the results?

Yes

Is the language acceptable?

Yes

Do you have any ethical concerns with this paper?

No

Have you any concerns about statistical analyses in this paper?

No

Recommendation?

Major revision is needed (please make suggestions in comments)

Comments to the Author(s)

Thank you for the opportunity to review the manuscript titled "Combining epidemiological and ecological methods to quantify environmental and social effects on E. coli transmission.". The authors explored the importance of environmental conditions and social relationships at influencing E. coli transmission using contact and social networks and transmission experiments in cattle kept in an outdoor pen. Cattle were examined in the summers of 2017 and 2018 and results were compared across years. In 2017 environmental transmission was most important and in 2018 social interactions were more important. Because there were differences in precipitation between years, the authors conclude that precipitation was a leading factor. More broadly the authors conclude that for an enteric microbe like E. coli, transmission pathways can vary over time.

This is an interesting study that provides useful information on how enteric microbes could be transmitted in domestic animal populations, and possibly to free-ranging populations. However, I have several concerns about the assumptions made by the authors, the lack of information provided in some instances, and the limited information provided about the importance of the study, who the audience is, and what the implications are of their findings both in terms of what future studies should focus on and what interventions should be taken by practitioners and managers. Below is a list of the major comments and further down are line by line suggestions. Overall, I think this study is important but requires major revisions.

General comments:

My biggest concern with this study is that the authors are exploring different pathways of transmission by looking at E. coli but it is unclear whether multiple strains are examined or just one, if one, which one and why. The authors mention in the introduction that they targeted 'a traceable commensal E. coli strain' but the methods are too vague to understand how strains were differentiated and the results present no information about the types of strains found. It is known that E. coli strains can differ in transmission potential and thus pathways. Therefore, providing more background information on the target strain(s) is needed. Further, strain type might also explain the difference between years. If the authors did not distinguish between different strains, I suggest toning down the study conclusions and adding the lack of differentiation of strains as a limitation in the discussion. Likewise, examining presence/absence of E. coli is not as informative as comparing CFU. Did the authors explore this? It is unclear currently. If not, suggest adding it as a future direction.

Secondly, one of the main objectives of this study is to explore how environmental conditions influence transmission. However, the only abiotic factor examined is precipitation. Yet, several other factors are known to influence E. coli persistence in the environment, such as temperature and soil moisture. At minimum the authors should examine the effect of temperature, and discuss the importance of other abiotic factors.

Finally, I found the authors description of the study limitations very good, however suggest using these limitations to provide guidance for future studies. For example, the authors mention that they only looked at one season. The authors could provide more information/discussion on what we would expect in other systems and why its important for other studies to make comparisons or focus on another season. Similarly, the authors acknowledged that they were not able to identify specific social relationships. It would be good to discuss what the different social relationships would be and why its important to differentiate them for E. coli transmission

Specific comments:

Line 31: Suggest providing examples of different environmental conditions in brackets to provide context.

Line 32: Same as above. What kind of social relationships? Suggest providing examples in brackets.

Line 32: Suggest defining an “E. coli outbreak”. I would avoid talking about ‘outbreak’ since the vast majority of E. coli are commensal. A way around this, would be to clearly state in the introduction that the authors are using commensal E. coli as a model for other pathogenic and non-pathogenic microbes.

Line 37-38: OK, but how can these findings be used to inform interventions? The authors mention earlier in the abstract that that is the purpose of the study.

Lines 46-56: Suggest not zooming into to enteric microparasites so soon in the introduction because the authors subsequently zoom out to social and contact networks while referencing papers that do not work specifically on enteric microparasites. Suggest moving this section further down when introducing the system.

Line 79-83: OK, but some of the papers referenced are exploring transmission in free-ranging, wild animals. Suggest the authors state explicitly that they will be adding on to this previous work by testing different pathways under controlled conditions in domestic animal systems. Instead of pointing out gaps in the previous studies, suggest highlighting that the authors study is filling an important gap that cannot easily be tested in free-ranging systems.

Line 87: An alternative approach to what? Unclear which component the authors are referring to. Suggest clarifying.

Line 93-94: Given that this approach has been used for almost two decades now, referencing a 2019 paper seems inappropriate and disregards the important work that has been done in previous years. Suggest referencing earlier work. More broadly, there seems to be a pattern in the introduction where the authors reference very recent papers that were not the first to come up with the point made. Suggest going through each statement and making sure the most appropriate studies are being referenced.

Line 100: Suggest providing examples of these social behaviors.

Line 103: How many? Suggest being specific here.

Line 104: Which strain and what was the reasoning for selecting it? Has it been used in other work?

Line 117: Why was this timeframe chosen? Related to E. coli biology? Suggest clarifying the need to examine two years.

Line 121-122: Why is that important? Suggest providing more information.

Line 126: How was the random selection done?

Line 127: How was this cut-off chosen? Was it based on previous work?

Line 130-131: How was this preliminary analysis conducted? Could provide the methods and results in the form of supplementary materials.

Line 133-135: This is currently coming out of nowhere. Suggest providing more information on the reasoning for doing this.

Line 136: How was the inoculated strain detected?

Line 144: Suggest providing sensitivity and specificity also.

Line 153-158: This is the first-time precipitation data is mentioned. Suggest introducing precipitation in the introduction. The authors also need to clarify the relevance for E. coli. Further, why were temperature data not also examined? And other environmental factors, like soil moisture?

Line 162: What was the reasoning behind using 3 days?

Contact and social network, and hazard modelling sections: Well described.

Line 242: How often did that happen? Suggest providing specific numbers.

Line 362: Suggest clarifying what “long periods of time” means. Can you be more specific?

Line 364-365: An alternative explanation is that there were other factors that the authors did not account for that were influencing the results. Suggest providing examples of other factors not controlled for in this study that could have influenced these findings based on previous studies.

Line 366-367: Is there an incline in the pen? Otherwise I would not expect this to be the case. Suggest clarifying a bit more.

Lines 368-369: One of the references suggests that animals will favor concrete over soil. With such high-resolution movement data, the authors should be able to determine time spent on concrete vs. soil. Suggest clarifying this. Additionally, the authors should be able to determine the amount of movement vs. standing time, which could also help clarify this point. In this regard, suggest also comparing these differences between 2017 and 2018. Additionally, it was not totally clear to me whether individuals studied in 2017 were the same individuals studied in 2018? If different, this may also be influencing the findings and should be acknowledged.

Lines 406-407: It was not clear from the introduction what these findings would be used for. Thus, listing this as a limitation is good but it reinforces the point that the authors need to clearly state who these findings are directed to. It seems a stretch to try and pitch this work to free-ranging animal systems.

Lines 408-409: It would be good to provide more information here. Which seasons have more shedding? And does cattle behavior differ across seasons? If so, in which direction and what does this mean for your summer findings?

Lines 437-438: But the authors don't say how their findings help guide controlling transmission. Figures and tables: Clear and informative.

Decision letter (RSOS-210328.R0)

Dear Mr Farthing

The Editors assigned to your paper RSOS-210328 "Combining epidemiological and ecological methods to quantify environmental and social effects on E. coli transmission" have now received comments from reviewers and would like you to revise the paper in accordance with the reviewer comments and any comments from the Editors. Please note this decision does not guarantee eventual acceptance.

Please submit your revised manuscript and required files (see below) no later than 21 days from today's (ie 17-Jun-2021) date. Note: the ScholarOne system will 'lock' if submission of the revision is attempted 21 or more days after the deadline. If you do not think you will be able to meet this deadline please contact the editorial office immediately.

on behalf of Professor Joshua Ross (Associate Editor) and Pete Smith (Subject Editor)
openscience@royalsociety.org

Associate Editor Comments to Author (Professor Joshua Ross):

Both Reviewers have identified potential major issues that will require some work and time to correct. However, both Reviewers also believe the work to be of much value if those issues can be addressed. For this reason, I have recommended Major Revision.

Reviewer comments to Author:

Reviewer: 1

Comments to the Author(s)

This is well written analysis of an ambitious and exciting attempt to quantify the contribution of physical and social network structure on the transmission of E. coli between cattle under realistic production conditions. High resolution spatial tracking data is used to characterise dynamic contact and social networks within two experimental groups of cattle. This network data is then used to attempt to tease apart the direct and indirect contributions to the natural transmission of E.coli (following experimental inoculation of a small number of seeder animals). The manuscript is supported by an exemplary commitment to open science, with raw data, processed data sets and code made available for review which greatly assisted this process.

However, I have some major concerns about both the formulation of the hazard models that provide the central results and the (lack) of robust model checking which I suspect means that models which have failed to converge have been retained for model selection (and in the case of the 2017 data set selected as the best fitting model).

The pairwise accelerated failure time regression models as implemented in the transtat package are an astute and appropriate framework with which to model this data and address the authors central research question. However, I am concerned about the biological interpretation of some of the covariate terms given how the regression terms are both described in the manuscript (equation 3) and implemented in the associated online code. Equation 3 implies that environmental co-variates (Ij,penSample, precip, social) contribute an equal (multiplicative) factor to the risk of transmission between each susceptible-infected pair within the herd. As such, are not each of these environmental variables being implicitly weighted by the instantaneous number of infectious-susceptible pairs at each time point? My understanding is that the number of infectious individuals at each time is already implicitly accounted for by the baseline hazard

(intercept term)? By adding I_j as an separate covariate is this term not effectively introducing a non-linear hazard response?

In relation to this issue I also note that the preprint by Sharker & Kenah (2019) cited by the authors describes a mixed modelling approach to include both "internal transmission" (as represented by the dyadic interactions modelled here) and "external transmission" terms (including environmental covariates). This would appear to be ideal for your use case, but is not what you describe in the manuscript or apparently how your code is functioning.

If my understanding is correct and the intercept and I_j terms are effectively modelling the same effect, one might expect this to lead to identifiability or convergence issues for the MLE procedure. Examining the best fit model for the 2017 data set does indeed suggest a convergence issue as the variance estimate for one of the model parameters is negative. The online source code suggests this may be a bug with the reporting of the transtat package leading to missing confidence intervals. However, I would note that the confidence intervals can be calculated using the slower likelihood ratio method (rather than the Wald approximation) and are infinite, suggesting that in these cases the parameter estimates are singular rather than this being a reporting (or rounding issue). There is further evidence of convergence issues for these models given that different optimisation routines return wildly different parameter estimates (and associated log-likelihoods).

These symptoms, I would suggest, are most likely to mean that the models demonstrating these issues are misspecified or poorly fit the data. The log-likelihood values presented only give a relative measure of fit based on comparison between the alternative models explored and do not provide a quantitative measure of the extent to which the fitted models can explain the original data set. The exceptionally high (weighted mean) hazard rates presented in Figure 4 for the 2017 models seem biologically implausible given the dramatically smaller number of colonisation events in the first two weeks for the 2017 experiment compared to 2018. This again suggests either convergence or fit issues to the 2017 data, raising the possibility that the hazard models simply can't explain the observed data for 2017 given the available covariates. In addition to plotting the hazard rates derived from the models it would be useful to see the predicted survival curves from the fitted models.

In conclusion, this is an exciting and novel data set and application of a cutting edge statistical analysis that could be a terrific contribution to the field if the concerns above can be addressed with further evidence of appropriate model checks.

Minor comments

- 1) I would consider the derived social and contact networks in the groups to be of intrinsic interest themselves and was disappointed not to have seen them compared (or at least visualised!). The procedure to identify social links is a pragmatic approach that makes sense algorithmically. But are the network properties of the inferred social networks consistent with those found in observational studies of cattle behaviour? Also, how do the properties of the social network differ from the contact networks?
- 2) Figure 3, there are no uncertainty intervals plotted on the survival curve
- 3) The difference in the temporal sampling of animals positions in the two years of the study is described in multiple places in the manuscript (and in detail in supplemental information). However, the reason for this difference is not given. Was this an intentional change in the sampling frequency for operational or logistic reasons, or was it the result of losing a greater number of records due to filtering?

- 4) With respect to the difference in sampling frequency between the two years, would be worth considering doing a sensitivity analysis downsampling the 2018 data to show how/if the reduced resolution impacts on the measured contact and social networks
- 5) Line 245, I am a little confused by the implication that the 2017 data set was terminated early due to two weeks in a row with 0 positive fecal samples. Should this not be 0 negative fecal samples, or was *E. coli* eliminated from this herd? Other figures and data suggest not.

Reviewer: 2

Comments to the Author(s)

Thank you for the opportunity to review the manuscript titled "Combining epidemiological and ecological methods to quantify environmental and social effects on *E. coli* transmission.". The authors explored the importance of environmental conditions and social relationships at influencing *E. coli* transmission using contact and social networks and transmission experiments in cattle kept in an outdoor pen. Cattle were examined in the summers of 2017 and 2018 and results were compared across years. In 2017 environmental transmission was most important and in 2018 social interactions were more important. Because there were differences in precipitation between years, the authors conclude that precipitation was a leading factor. More broadly the authors conclude that for an enteric microbe like *E. coli*, transmission pathways can vary over time.

This is an interesting study that provides useful information on how enteric microbes could be transmitted in domestic animal populations, and possibly to free-ranging populations. However, I have several concerns about the assumptions made by the authors, the lack of information provided in some instances, and the limited information provided about the importance of the study, who the audience is, and what the implications are of their findings both in terms of what future studies should focus on and what interventions should be taken by practitioners and managers. Below is a list of the major comments and further down are line by line suggestions. Overall, I think this study is important but requires major revisions.

General comments:

My biggest concern with this study is that the authors are exploring different pathways of transmission by looking at *E. coli* but it is unclear whether multiple strains are examined or just one, if one, which one and why. The authors mention in the introduction that they targeted 'a traceable commensal *E. coli* strain' but the methods are too vague to understand how strains were differentiated and the results present no information about the types of strains found. It is known that *E. coli* strains can differ in transmission potential and thus pathways. Therefore, providing more background information on the target strain(s) is needed. Further, strain type might also explain the difference between years. If the authors did not distinguish between different strains, I suggest toning down the study conclusions and adding the lack of differentiation of strains as a limitation in the discussion. Likewise, examining presence/absence of *E. coli* is not as informative as comparing CFU. Did the authors explore this? It is unclear currently. If not, suggest adding it as a future direction.

Secondly, one of the main objectives of this study is to explore how environmental conditions influence transmission. However, the only abiotic factor examined is precipitation. Yet, several other factors are known to influence *E. coli* persistence in the environment, such as temperature and soil moisture. At minimum the authors should examine the effect of temperature, and discuss the importance of other abiotic factors.

Finally, I found the authors description of the study limitations very good, however suggest using these limitations to provide guidance for future studies. For example, the authors mention that they only looked at one season. The authors could provide more information/discussion on what we would expect in other systems and why its important for other studies to make comparisons or focus on another season. Similarly, the authors acknowledged that they were not able to identify specific social relationships. It would be good to discuss what the different social relationships would be and why its important to differentiate them for *E. coli* transmission

Specific comments:

Line 31: Suggest providing examples of different environmental conditions in brackets to provide context.

Line 32: Same as above. What kind of social relationships? Suggest providing examples in brackets.

Line 32: Suggest defining an “E. coli outbreak”. I would avoid talking about ‘outbreak’ since the vast majority of E. coli are commensal. A way around this, would be to clearly state in the introduction that the authors are using commensal E. coli as a model for other pathogenic and non-pathogenic microbes.

Line 37-38: OK, but how can these findings be used to inform interventions? The authors mention earlier in the abstract that that is the purpose of the study.

Lines 46-56: Suggest not zooming into to enteric microparasites so soon in the introduction because the authors subsequently zoom out to social and contact networks while referencing papers that do not work specifically on enteric microparasites. Suggest moving this section further down when introducing the system.

Line 79-83: OK, but some of the papers referenced are exploring transmission in free-ranging, wild animals. Suggest the authors state explicitly that they will be adding on to this previous work by testing different pathways under controlled conditions in domestic animal systems. Instead of pointing out gaps in the previous studies, suggest highlighting that the authors study is filling an important gap that cannot easily be tested in free-ranging systems.

Line 87: An alternative approach to what? Unclear which component the authors are referring to. Suggest clarifying.

Line 93-94: Given that this approach has been used for almost two decades now, referencing a 2019 paper seems inappropriate and disregards the important work that has been done in previous years. Suggest referencing earlier work. More broadly, there seems to be a pattern in the introduction where the authors reference very recent papers that were not the first to come up with the point made. Suggest going through each statement and making sure the most appropriate studies are being referenced.

Line 100: Suggest providing examples of these social behaviors.

Line 103: How many? Suggest being specific here.

Line 104: Which strain and what was the reasoning for selecting it? Has it been used in other work?

Line 117: Why was this timeframe chosen? Related to E. coli biology? Suggest clarifying the need to examine two years.

Line 121-122: Why is that important? Suggest providing more information.

Line 126: How was the random selection done?

Line 127: How was this cut-off chosen? Was it based on previous work?

Line 130-131: How was this preliminary analysis conducted? Could provide the methods and results in the form of supplementary materials.

Line 133-135: This is currently coming out of nowhere. Suggest providing more information on the reasoning for doing this.

Line 136: How was the inoculated strain detected?

Line 144: Suggest providing sensitivity and specificity also.

Line 153-158: This is the first-time precipitation data is mentioned. Suggest introducing precipitation in the introduction. The authors also need to clarify the relevance for E. coli. Further, why were temperature data not also examined? And other environmental factors, like soil moisture?

Line 162: What was the reasoning behind using 3 days?

Contact and social network, and hazard modelling sections: Well described.

Line 242: How often did that happen? Suggest providing specific numbers.

Line 362: Suggest clarifying what “long periods of time” means. Can you be more specific?

Line 364-365: An alternative explanation is that there were other factors that the authors did not account for that were influencing the results. Suggest providing examples of other factors not controlled for in this study that could have influenced these findings based on previous studies.

Line 366-367: Is there an incline in the pen? Otherwise I would not expect this to be the case.

Suggest clarifying a bit more.

Lines 368-369: One of the references suggests that animals will favor concrete over soil. With such high-resolution movement data, the authors should be able to determine time spent on concrete vs. soil. Suggest clarifying this. Additionally, the authors should be able to determine the amount of movement vs. standing time, which could also help clarify this point. In this regard, suggest also comparing these differences between 2017 and 2018. Additionally, it was not totally clear to me whether individuals studied in 2017 were the same individuals studied in 2018? If different, this may also be influencing the findings and should be acknowledged.

Lines 406-407: It was not clear from the introduction what these findings would be used for. Thus, listing this as a limitation is good but it reinforces the point that the authors need to clearly state who these findings are directed to. It seems a stretch to try and pitch this work to free-ranging animal systems.

Lines 408-409: It would be good to provide more information here. Which seasons have more shedding? And does cattle behavior differ across seasons? If so, in which direction and what does this mean for your summer findings?

Lines 437-438: But the authors don't say how their findings help guide controlling transmission.

Figures and tables: Clear and informative.

===PREPARING YOUR MANUSCRIPT===

===PREPARING YOUR REVISION IN SCHOLARONE===

Author's Response to Decision Letter for (RSOS-210328.R0)

See Appendix A.

RSOS-210328.R1 (Revision)

Review form: Reviewer 1

Is the manuscript scientifically sound in its present form?

No

Are the interpretations and conclusions justified by the results?

No

Is the language acceptable?

Yes

Do you have any ethical concerns with this paper?

No

Have you any concerns about statistical analyses in this paper?

Yes

Recommendation?

Accept with minor revision (please list in comments)

Comments to the Author(s)

The authors removal of environmental co-variates addresses my main concerns with the previous submission. However, the new presentation of predictive performance of the new best-fit models raises further questions. This appears that it may be due to a minor mistake, but further review of this work depends on clarification of this key point.

Specifically, I am concerned that in Figure 2 (panels b,c) the authors have made the wrong comparison between the empirical and predicted survival probabilities. The panel claims to compare the weekly survival probability, but the trend and shape of the model predictions look more like cumulative survival probabilities (as plotted in panel a). Indeed, agreement between the model predictions and the empirical cumulative survival probability is much better and in line with the narrative in the discussion - with qualitatively good agreement for 2018 and poorer agreement for 2017 (driven by the exceptionally high observed transmission at the first time point).

If the current version of Figure 2 is correct the predictive performance of the fitted models is not even qualitatively correct. Such a systematic lack of fit would make interpretation of the

estimated effects from the fitted models highly problematic - statistical significance is meaningless if the model cannot at least approximate the data. As presented this discrepancy is not "relatively poor" (Line 488), but completely unrepresentative.

Assuming the predictive checks have been plotted incorrectly I would still disagree with the authors conclusion that "Our results suggest that the number of close-proximity contacts and social relationships with shedding individuals are poor predictors for E. coli colonization in susceptible feedlot cattle overall, but can influence colonization rates." The former is clearly supported, but I would argue that the latter can at best not be ruled out rather than supported. For both years the additional information from the networks does not increase the predictive accuracy compared to the null model. The difference in AIC are small, which again I would argue indicates that we cannot rule out that the network structure impacts transmission risk there is no evidence to support it in this data. (i.e. the effect may be true but too small to be estimated in this study).

This "null" result is absolutely still important on it's own given that our baseline assumption for infectious disease transmission is that social and physical contacts will increase the risk of transmission. The difficulty we have in confirming this intuitive assumption is for me a key open research question. For this system where transmission is mediated through the environment, the "null" result makes sense and I agree that using the environmental data could help to improve predictive ability. While I appreciate that developing new statistical methodology is out of the scope of this paper, I am a little disappointed that the environmental data was dropped completely. A null analysis comparing the predictive power of a model only trained using environmental co-variates should be straightforward and very much justified by the lack of importance of network structure suggested by these results.

Minor comments

Response to R1.11. I would respectively disagree that the difference in sparsity between the two years is outside of the scope of this paper. If you wish to compare between the two years, then I would consider this an important (albeit supplemental) sensitivity analysis at least to show that downsampling to the same level in both datasets has no impact on results. (Given the lack of importance of the networks I would absolutely expect this to be the case, and absolutely would not consider a systematic exploration to be necessary).

Line 240: The benefit of sparsity in location reporting during the day is presumably a smaller data set to work with? The latter motivation - i.e. that animals will be more active socially at these times is presumably the more important factor. When reading this I at first thought measurements at nighttime were more sparse, hence a reason to discount them. A little extra context (or simply focusing on the argument to focus on socially active hours) might make this clearer.

Line 432: Unless the github repository is to be archived/make read only after publication would be better to provide this in supplemental information (or simply rephrase to say that code can be used to recreate the full table)

Review form: Reviewer 2

Is the manuscript scientifically sound in its present form?

Yes

Are the interpretations and conclusions justified by the results?

Yes

Is the language acceptable?

Yes

Do you have any ethical concerns with this paper?

No

Have you any concerns about statistical analyses in this paper?

No

Recommendation?

Accept as is

Comments to the Author(s)

Thank you for carefully and thoroughly addressing the issues raised in the previous review. No additional comments.

Decision letter (RSOS-210328.R1)

Dear Mr Farthing

On behalf of the Editors, we are pleased to inform you that your Manuscript RSOS-210328.R1 "Combining epidemiological and ecological methods to quantify social effects on E. coli transmission" has been accepted for publication in Royal Society Open Science subject to minor revision in accordance with the referees' reports. Please find the referees' comments along with any feedback from the Editors below my signature.

Please submit your revised manuscript and required files (see below) no later than 7 days from today's (ie 16-Aug-2021) date. Note: the ScholarOne system will 'lock' if submission of the revision is attempted 7 or more days after the deadline. If you do not think you will be able to meet this deadline please contact the editorial office immediately.

on behalf of Professor Joshua Ross (Associate Editor) and Pete Smith (Subject Editor)
openscience@royalsociety.org

Reviewer comments to Author:

Reviewer: 1

Comments to the Author(s)

The authors removal of environmental co-variates addresses my main concerns with the previous submission. However, the new presentation of predictive performance of the new best-fit models raises further questions. This appears that it may be due to a minor mistake, but further review of this work depends on clarification of this key point.

Specifically, I am concerned that in Figure 2 (panels b,c) the authors have made the wrong comparison between the empirical and predicted survival probabilities. The panel claims to compare the weekly survival probability, but the trend and shape of the model predictions look more like cumulative survival probabilities (as plotted in panel a). Indeed, agreement between the model predictions and the empirical cumulative survival probability is much better and in line with the narrative in the discussion - with qualitatively good agreement for 2018 and poorer agreement for 2017 (driven by the exceptionally high observed transmission at the first time point).

If the current version of Figure 2 is correct the predictive performance of the fitted models is not even qualitatively correct. Such a systematic lack of fit would make interpretation of the estimated effects from the fitted models highly problematic - statistical significance is meaningless if the model cannot at least approximate the data. As presented this discrepancy is not "relatively poor" (Line 488), but completely unrepresentative.

Assuming the predictive checks have been plotted incorrectly I would still disagree with the authors conclusion that "Our results suggest that the number of close-proximity contacts and social relationships with shedding individuals are poor predictors for E. coli colonization in susceptible feedlot cattle overall, but can influence colonization rates." The former is clearly supported, but I would argue that the latter can at best not be ruled out rather than supported. For both years the additional information from the networks does not increase the predictive accuracy compared to the null model. The difference in AIC are small, which again I would argue indicates that we cannot rule out that the network structure impacts transmission risk there is no evidence to support it in this data. (i.e. the effect may be true but too small to be estimated in this study).

This "null" result is absolutely still important on it's own given that our baseline assumption for infectious disease transmission is that social and physical contacts will increase the risk of transmission. The difficulty we have in confirming this intuitive assumption is for me a key open research question. For this system where transmission is mediated through the environment, the "null" result makes sense and I agree that using the environmental data could help to improve predictive ability. While I appreciate that developing new statistical methodology is out of the scope of this paper, I am a little disappointed that the environmental data was dropped completely. A null analysis comparing the predictive power of a model only trained using environmental co-variates should be straightforward and very much justified by the lack of importance of network structure suggested by these results.

Minor comments

Response to R1.11. I would respectively disagree that the difference in sparsity between the two years is outside of the scope of this paper. If you wish to compare between the two years, then I would consider this an important (albeit supplemental) sensitivity analysis at least to show that downsampling to the same level in both datasets has no impact on results. (Given the lack of importance of the networks I would absolutely expect this to be the case, and absolutely would not consider a systematic exploration to be necessary).

Line 240: The benefit of sparsity in location reporting during the day is presumably a smaller data set to work with? The latter motivation - i.e. that animals will be more active socially at these times is presumably the more important factor. When reading this I at first thought measurements at nighttime were more sparse, hence a reason to discount them. A little extra context (or simply focusing on the argument to focus on socially active hours) might make this clearer.

Line 432: Unless the github repository is to be archived/make read only after publication would be better to provide this in supplemental information (or simply rephrase to say that code can be used to recreate the full table)

Reviewer: 2

Comments to the Author(s)

Thank you for carefully and thoroughly addressing the issues raised in the previous review. No additional comments.

===PREPARING YOUR MANUSCRIPT===

If you have been asked to revise the written English in your submission as a condition of publication, you must do so, and you are expected to provide evidence that you have received language editing support. The journal would prefer that you use a professional language editing service and provide a certificate of editing, but a signed letter from a colleague who is a native speaker of English is acceptable. Note the journal has arranged a number of discounts for authors

using professional language editing services
(<https://royalsociety.org/journals/authors/benefits/language-editing/>).

===PREPARING YOUR REVISION IN SCHOLARONE===

-- If you have uploaded ESM files, please ensure you follow the guidance at <https://royalsociety.org/journals/authors/author-guidelines/#supplementary-material> to include a suitable title and informative caption. An example of appropriate titling and captioning may be found at https://figshare.com/articles/Table_S2_from_Is_there_a_trade-

off_between_peak_performance_and_performance_breadth_across_temperatures_for_aerobic_sc
ope_in_teleost_fishes_/3843624.

Author's Response to Decision Letter for (RSOS-210328.R1)

See Appendix B.

Decision letter (RSOS-210328.R2)

Dear Mr Farthing,

I am pleased to inform you that your manuscript entitled "Combining epidemiological and ecological methods to quantify social effects on E. coli transmission" is now accepted for publication in Royal Society Open Science.

on behalf of Professor Joshua Ross (Associate Editor) and Pete Smith (Subject Editor)
openscience@royalsociety.org

Appendix A

Dear Editor,

Please consider our revised manuscript, “Combining epidemiological and ecological methods to quantify social effects on *E. coli* transmission,” for publication in *Royal Society Open Science*.

We sincerely appreciated the editorial and reviewer comments we received, and have incorporated them into our manuscript to create a much stronger paper. We addressed the major concerns that reviewers indicated and have rewritten portions of the manuscript to increase clarity. Most significantly, we have corrected our use of pairwise accelerated failure time modeling procedures by excluding population-stable environmental covariates from models, as recommended by Reviewer 1. We have also clarified portions of our Methods section to ensure readers understand our how our transmission experiments were carried out, and reported yearly contact- and social-network metrics.

Our Results and Discussion sections have necessarily changed due to the removal of environmental covariates from our models, as we can no longer comment on environment-related predictors of *E. coli* transmission. Despite the reduced scope of our analyses however, our findings remain largely unchanged from the previously-submitted draft. That is to say, we continue to demonstrate that transmission experiments can be combined with real-time location data collection and processing procedures to create an effective framework for quantifying sociobehavioral effects on microparasite transmission. We also demonstrate how ecological methods for identifying sociality in animal populations can be adapted to characterize specific social relationships between individuals. Finally, we show that our sociobehavioral covariates were generally poor predictors of *E. coli* colonization in our study populations, but they can have significant ($p \leq 0.05$) effects on colonization rates. Significant effects on transmission however, were not consistent between similar populations, which highlights the potential variability inherent in these microparasite-host systems.

Due to the wide-ranging applicability of our methods, and our contributions to the growing body of literature on sociobehavioral drivers of enteric microparasite transmission, we continue to feel that our work is well suited for publication in *Royal Society Open Science*. We include a point-by-point response to editorial and reviewer comments below. Editor and reviewer comments are presented in black, and our responses are in blue.

Thank you again for considering our revised manuscript.

Sincerely,

Trevor Farthing

Trevor Farthing, M.S.
Graduate Research Assistant
North Carolina State University
Raleigh, NC 27695
tsfarthi@ncsu.edu

E1) Associate Editor Comments to Author (Professor Joshua Ross)

E1.1) Both Reviewers have identified potential major issues that will require some work and time to correct. However, both Reviewers also believe the work to be of much value if those issues can be addressed. For this reason, I have recommended Major Revision.

Understood. We have addressed all reviewer concerns in order (see below for detailed descriptions). Our responses are given in blue. A *Literature Cited* section is provided at the end of the document, and contains references for in-text citations below.

R1) Reviewer 1 Comments

R1.1) This is well written analysis of an ambitious and exciting attempt to quantify the contribution of physical and social network structure on the transmission of E. coli between cattle under realistic production conditions. High resolution spatial tracking data is used to characterise dynamic contact and social networks within two experimental groups of cattle. This network data is then used to attempt to tease apart the direct and indirect contributions to the natural transmission of E.coli (following experimental inoculation of a small number of seeder animals). The manuscript is supported by an exemplary commitment to open science, with raw data, processed data sets and code made available for review which greatly assisted this process.

Thank you. We strive to make our research processes as clear and accessible as possible.

R1.2) However, I have some major concerns about both the formulation of the hazard models that provide the central results and the (lack) of robust model checking which I suspect means that models which have failed to converge have been retained for model selection (and in the case of the 2017 data set selected as the best fitting model).

After reviewing your comments and literature on the statistical methods we employed, we agree with you. We have removed all candidate models from our model sets (both 2017 and 2018) that include environmental covariates. Models without these problem covariates converge properly.

R1.3) The pairwise accelerated failure time regression models as implemented in the transtat package are an astute and appropriate framework with which to model this data and address the authors central research question. However, I am concerned about the biological interpretation of some of the covariate terms given how the regression terms are both described in the manuscript (equation 3) and implemented in the associated online code. Equation 3 implies that environmental co-variates (I_j, penSample, precip, social) contribute an equal (multiplicative) factor to the risk of transmission between each susceptible-infected pair within the herd. As such, are not each of these environmental variables being implicitly weighted by the instantaneous number of infectious-susceptible pairs at each time point? My understanding is that the number of infectious individuals at each time is already implicitly accounted for by the baseline hazard (intercept term)? By adding I_j as an separate covariate is this term not effectively introducing a non-linear hazard response?

As noted in our response to your previous comment, we agree with your assessment and have made appropriate changes throughout our manuscript. Specifically, we have removed all candidate models from our model sets (both 2017 and 2018) that include environmental covariates, and rewritten the Methods, Results, and Discussion sections to reflect this change. Changes to the Methods section are relatively minor, but large portions Results and Discussion sections have been rewritten to reflect the updated “best-fitting” model information. Thank you for bringing this to our attention.

R1.4) In relation to this issue I also note that the preprint by Sharker & Kenah (2019) cited by the authors describes a mixed modelling approach to include both "internal transmission" (as represented by the dyadic interactions modelled here) and "external transmission" terms (including environmental covariates). This would appear to be ideal for your use case, but is not what you describe in the manuscript or apparently how your code is functioning.

We thoroughly considered using the methods presented by Sharker & Kenah (2019), but after careful review determined that they were inappropriate for modeling environment-mediated *E. coli* colonization rates in our study system. This is because their methods assume that effects of internal (i.e., direct transmission from infectious individuals to susceptible ones) and external transmission (i.e., environment-mediated transmission in our system) on hazard rates are independent. This assumption is violated in our study system because we expect environment-mediated transmission risk to fluctuate as colonized individuals shed *E. coli* colony-forming units. We now highlight the need for further statistical method development in this area in lines L415-L426 of our manuscript.

R1.5) If my understanding is correct and the intercept and I_j terms are effectively modelling the same effect, one might expect this to lead to identifiability or convergence issues for the MLE procedure. Examining the best fit model for the 2017 data set does indeed suggest a convergence issue as the variance estimate for one of the model parameters is negative. The online source code suggests this may be a bug with the reporting of the transtat package leading to missing confidence intervals. However, I would note that the confidence intervals can be calculated using the slower likelihood ratio method (rather than the Wald approximation) and are infinite, suggesting that in these cases the parameter estimates are singular rather than this being a reporting (or rounding issue). There is further evidence of convergence issues for these models given that different optimisation routines return wildly different parameter estimates (and associated log-likelihoods).

Again, we agree with your assessment and have made appropriate changes throughout our manuscript. (see our responses to comments R1.2 and R1.3 for more details).

R1.6) These symptoms, I would suggest, are most likely to mean that the models demonstrating these issues are misspecified or poorly fit the data. The log-likelihood values presented only give a relative measure of fit based on comparison between the alternative models explored and do not provide a quantitative measure of the extent to which the fitted models can explain the original data set. The exceptionally high (weighted mean) hazard rates presented in Figure 4 for the 2017 models seem biologically implausible given the dramatically smaller number of colonisation events in the first two weeks for the 2017 experiment compared to 2018. This again

suggests either convergence or fit issues to the 2017 data, raising the possibility that the hazard models simply can't explain the observed data for 2017 given the available covariates. In addition to plotting the hazard rates derived from the models it would be useful to see the predicted survival curves from the fitted models.

Agreed. We have replaced estimates of AIC-weighted mean weekly per-capita pairwise hazard estimates with estimates of weekly AIC-weighted survival rates as suggested (now shown in Figure 2b-c). By plotting these survival estimates together with empirical weekly survival probabilities observed in each study period (*see* Figure 2b-c) we were able to comment on our models' abilities to predict empirical survival outcomes (*see* L407-L410).

R1.7) In conclusion, this is an exciting and novel data set and application of a cutting edge statistical analysis that could be a terrific contribution to the field if the concerns above can be addressed with further evidence of appropriate model checks.

Thank you for your thorough exploration of our work. We feel we have completely addressed your comments, and the manuscript is much stronger as a result.

R1.8) I would consider the derived social and contact networks in the groups to be of intrinsic interest themselves and was disappointed not to have seen them compared (or at least visualised!). The procedure to identify social links is a pragmatic approach that makes sense algorithmically. But are the network properties of the inferred social networks consistent with those found in observational studies of cattle behaviour? Also, how do the properties of the social network differ from the contact networks?

Weekly contact and social network metrics are now described in Tables 2 and 3 to allow for direct comparisons and review of network structures between years. Visualizations would not be very informative due to the density of these networks. To answer your question about the biological significance of our social networks, there is actually not much available information on steer social structure in feedlot settings. We know that cattle exhibit sociality and that rates of interaction with others varies between individuals (Rocha et al. 2020). In this regard, our networks are in line with previous findings. However, previous work utilizing contact data to describe cattle interactions have primarily focused on interactions between cows or cows and their calves (Swain & Bishop-Hurley 2007; Kleinlützum *et al.* 2013; Rocha et al. 2020). We expect social dynamics between confined steers to be quite different. Furthermore, we are the first to adapt the Spiegel et al. (2016) methodology to estimate social interactions in any animal species. For these reasons, it is difficult to directly compare our networks to previous works.

R1.9) Figure 3, there are no uncertainty intervals plotted on the survival curve.

This figure (formerly Figure 3, now Figure 2a) shows empirical survival counts observed each week. It does not illustrate statistical model predictions, and as such does not require uncertainty or confidence intervals. The Figure 2 caption now explicitly describes this figure as the "Proportion of 2017 and 2018 study populations observed remaining uncolonized by *E. coli* at weekly timesteps."

R1.10) The difference in the temporal sampling of animals positions in the two years of the study is described in multiple places in the manuscript (and in detail in supplemental information). However, the reason for this difference is not given. Was this an intentional change in the sampling frequency for operational or logistic reasons, or was it the result of losing a greater number of records due to filtering?

This was not an intentional change in our sampling regime or a result of an increased proportion of erroneous data points in 2018 relative to 2017. Rather, the real-time location system we used simply recorded fewer data points in 2018. The cause is ultimately unknown, but we suspect the issue arose from damaged or faulty real-time-location-system infrastructure associated with the feedlot pen in which the 2018 population was kept. This explanation is now stated explicitly in Appendix S1.

R1.11) With respect to the difference in sampling frequency between the two years, would be worth considering doing a sensitivity analysis downsampling the 2018 data to show how/if the reduced resolution impacts on the measured contact and social networks

This is a good point. We know from Dawson *et al.* (2019) that how point-location data are processed does influence contact network metrics and downstream transmission model outcomes. As we note in lines L354-L362, we do expect that social network disparities between years are likely a result of the relative data sparsity in the 2018 dataset. That said, evaluating effects of data-aggregation or -sparsity induced effects on social network metrics is outside the scope of this metric.

R1.12) Line 245, I am a little confused by the implication that the 2017 data set was terminated early due to two weeks in a row with 0 positive fecal samples. Should this not be 0 negative fecal samples, or was *E. coli* eliminated from this herd? Other figures and data suggest not.

We have rewritten L281-L287 to improve clarity. The lines now state: "Though we technically monitored individuals for 6 and 10 weeks in 2017 and 2018, respectively, because the pairwise AFT modeling methodology assumes that colonization risk to susceptible individuals only exists when shedding individuals are or will be present to propagate the epidemic (Kenah 2019), we can only use this procedure to model hazard rates up until the final timepoint when shedding individuals are observed. This occurred in weeks 5 and 10 of 2017 and 2018, respectively. Therefore, for 2017 we were only able to model hazard rates through week 5."

R2) Reviewer 2 Comments

R2.1) Thank you for the opportunity to review the manuscript titled "Combining epidemiological and ecological methods to quantify environmental and social effects on *E. coli* transmission.". The authors explored the importance of environmental conditions and social relationships at influencing *E. coli* transmission using contact and social networks and transmission experiments in cattle kept in an outdoor pen. Cattle were examined in the summers of 2017 and 2018 and results were compared across years. In 2017 environmental transmission was most important and in 2018 social interactions were more important. Because there were differences in precipitation

between years, the authors conclude that precipitation was a leading factor. More broadly the authors conclude that for an enteric microbe like *E. coli*, transmission pathways can vary over time.

Thank you for taking the time to review our paper. Our findings regarding environmental effects on colonization risk have been removed from the paper after careful consideration of Reviewer 1's comments.

R2.2) This is an interesting study that provides useful information on how enteric microbes could be transmitted in domestic animal populations, and possibly to free-ranging populations. However, I have several concerns about the assumptions made by the authors, the lack of information provided in some instances, and the limited information provided about the importance of the study, who the audience is, and what the implications are of their findings both in terms of what future studies should focus on and what interventions should be taken by practitioners and managers. Below is a list of the major comments and further down are line by line suggestions. Overall, I think this study is important but requires major revisions.

We address all of your specific concerns below.

R2.3) General comments:

My biggest concern with this study is that the authors are exploring different pathways of transmission by looking at *E. coli* but it is unclear whether multiple strains are examined or just one, if one, which one and why. The authors mention in the introduction that they targeted 'a traceable commensal *E. coli* strain' but the methods are too vague to understand how strains were differentiated and the results present no information about the types of strains found. It is known that *E. coli* strains can differ in transmission potential and thus pathways. Therefore, providing more background information on the target strain(s) is needed. Further, strain type might also explain the difference between years. If the authors did not distinguish between different strains, I suggest toning down the study conclusions and adding the lack of differentiation of strains as a limitation in the discussion. Likewise, examining presence/absence of *E. coli* is not as informative as comparing CFU. Did the authors explore this? It is unclear currently. If not, suggest adding it as a future direction.

Secondly, one of the main objectives of this study is to explore how environmental conditions influence transmission. However, the only abiotic factor examined is precipitation. Yet, several other factors are known to influence *E. coli* persistence in the environment, such as temperature and soil moisture. At minimum the authors should examine the effect of temperature, and discuss the importance of other abiotic factors.

Finally, I found the authors description of the study limitations very good, however suggest using these limitations to provide guidance for future studies. For example, the authors mention that they only looked at one season. The authors could provide more information/discussion on what we would expect in other systems and why its important for other studies to make comparisons or focus on another season. Similarly, the authors acknowledged that they were not able to identify specific social relationships. It would be good to discuss what the different social relationships would be and why its important to differentiate them for *E. coli* transmission

First, we only assessed the transmission of a single *E. coli* strain which was made resistant to nalidixic acid and rifampicin. Animals were screened prior to the transmission experiment to assure they were not colonized with bacteria carrying the dual resistance. Agar plates with nalidixic acid and rifampicin were used to identify the strain in the samples. This was made clearer throughout the manuscript. Presence/absence data for this strain is sufficiently informative for our purposes, as we sought to model the time to colonization with this single strain.

Second, after consideration of Reviewer 1's comments, the scope of our manuscript has been reduced and all mentions of environmental effects on colonization have been removed.

Finally, see below (R2.31) that we update a portion of our limitations section to provide more information about seasonal differences in *E. coli* dynamics and cattle behavior, as requested. Also, please note that we do give examples of potential social relationships that may exist within the study population. That said, we feel that our acknowledgement that the mentioned limitations *could* influence observed results is sufficient here. The fact is, that there are so many different factors that could affect colonization rates, any specific predictions would likely be inaccurate. As we have shown, even in our controlled studies we observed differences in significant ($p \leq 0.05$) predictors of colonization between 2017 and 2018 transmission experiments.

R2.4) Specific comments:

Line 31: Suggest providing examples of different environmental conditions in brackets to provide context.

As noted above, after consideration of Reviewer 1's comments, the scope of our manuscript has been reduced and all mentions of environmental effects on colonization have been removed.

R2.5) Line 32: Same as above. What kind of social relationships? Suggest providing examples in brackets.

L31-32 now specifies "estimated social affiliations and dyadic contact frequencies."

R2.6) Line 32: Suggest defining an "E. coli outbreak". I would avoid talking about 'outbreak' since the vast majority of *E. coli* are commensal. A way around this, would be to clearly state in the introduction that the authors are using commensal *E. coli* as a model for other pathogenic and non-pathogenic microbes.

In L28, we now specify that we "carried out commensal *E. coli* transmission experiments," and in L32 "*E. coli* outbreaks" has been replaced with "transmission experiments."

R2.7) Line 37-38: OK, but how can these findings be used to inform interventions? The authors mention earlier in the abstract that that is the purpose of the study.

Testing or otherwise evaluating effects of intervention strategies are outside the scope of this study. As social interactions were not a clear driver of transmission in this system, any consideration of these factors to inform interventions is not supported. L25-28 now read

“Characterizing the relative transmission risk attributable to host social relationships, and direct physical contact between individuals is paramount for understanding how microparasites like *E. coli* spread within affected communities and estimating colonization rates.”

R2.8) Lines 46-56: Suggest not zooming into to enteric microparasites so soon in the introduction because the authors subsequently zoom out to social and contact networks while referencing papers that do not work specifically on enteric microparasites. Suggest moving this section further down when introducing the system.

We have moved these lines to the beginning of the paragraph in which we discuss previous work on linking enteric pathogen transmission to contact and social interaction metrics. (*see* L59-L68).

R2.9) Line 79-83: OK, but some of the papers referenced are exploring transmission in free-ranging, wild animals. Suggest the authors state explicitly that they will be adding on to this previous work by testing different pathways under controlled conditions in domestic animal systems. Instead of pointing out gaps in the previous studies, suggest highlighting that the authors study is filling an important gap that cannot easily be tested in free-ranging systems.

The introduction has been changed to address the reviewer’s concerns, and in the subsequent paragraphs we do explicitly state what our study adds to the available literature.

R2.10) Line 87: An alternative approach to what? Unclear which component the authors are referring to. Suggest clarifying.

Done. The line now reads “The use of transmission experiments is an alternative approach for studying enteric pathogen transmission that does not share the limitations of studies examining the spread of endemic microparasites, as discussed above.”

R2.10) Line 93-94: Given that this approach has been used for almost two decades now, referencing a 2019 paper seems inappropriate and disregards the important work that has been done in previous years. Suggest referencing earlier work. More broadly, there seems to be a pattern in the introduction where the authors reference very recent papers that were not the first to come up with the point made. Suggest going through each statement and making sure the most appropriate studies are being referenced.

Additional citations have been added. Additionally, it is not our intention to discount early work on this subject matter. That said, throughout the manuscript we do prioritize references published within the previous decade to ensure that all information we present is as up-to-date as possible.

R2.11) Line 100: Suggest providing examples of these social behaviors.

Done.

R2.12) Line 103: How many? Suggest being specific here.

Done.

R2.13) Line 104: Which strain and what was the reasoning for selecting it? Has it been used in other work?

The *E. coli* strain used in our transmission experiments is explicitly described in section 2.1.2 (i.e., the “Longitudinal *E. coli* prevalence data” subsection of the Methods). It is a strain that we made resistant to nalidixic acid and rifampicin, allowing us to observe the presence or absence of colony-forming units in cultures treated with these antimicrobials.

R2.14) Line 117: Why was this timeframe chosen? Related to *E. coli* biology? Suggest clarifying the need to examine two years.

We performed experiments over multiple years to an effort to control for intra-year variation in *E. coli* colonization outcomes due to weather, different animals, etc. Transmission experiments were carried out over summer months to maximize the likelihood of observing ≥ 1 successful transmission event, as fecal shedding of *E. coli* by cattle, and *E. coli* prevalence in cattle populations are known to be relatively greater during warmer months (Hancock et al. 1997; Cobbold et al. 2004). This information has been added to Methods section in the manuscript.

R2.15) Line 121-122: Why is that important? Suggest providing more information.

Done. Clarifying information was added to the end of the sentence, which now reads: “The number of individuals included in each transmission experiment (i.e., $n = 70$) was intended to mimic stocking rates observed in U.S. concentrated animal feeding operations, ensuring that observed results reflect real-world colonization rates in these agricultural systems.”

R2.16) Line 126: How was the random selection done?

Using the random number generator in Excel (Microsoft Corporation 2016).

R2.17) Line 127: How was this cut-off chosen? Was it based on previous work?

We chose 5 days and 10^9 CFUs to maximize the probability of successfully establishing shedding in the inoculated calves.

R2.18) Line 130-131: How was this preliminary analysis conducted? Could provide the methods and results in the form of supplementary materials.

We inoculated a sufficient number of animals to assure that the outbreak would not fade out. The basic reproduction number for *E. coli* in cattle is ~ 4 ; therefore, the probability of fade-out when the initial number of infected is 2 to 5 is less than 10% (Diekmann & Heesterbeek, 2000).

R2.19) Line 133-135: This is currently coming out of nowhere. Suggest providing more information on the reasoning for doing this.

We have added detail and clarity to this section.

R2.20) Line 136: How was the inoculated strain detected?

We have added detail and clarity to this section.

R2.21) Line 144: Suggest providing sensitivity and specificity also.

The reported accuracy comes from the manufacturer's description of the real-time location system. This has been made clear in the document (*see* L153-154).

R2.22) Line 153-158: This is the first-time precipitation data is mentioned. Suggest introducing precipitation in the introduction. The authors also need to clarify the relevance for *E. coli*. Further, why were temperature data not also examined? And other environmental factors, like soil moisture?

As noted above, after consideration of Reviewer 1's comments, the scope of our manuscript has been reduced and all mentions of environmental effects on colonization have been removed.

R2.23) Line 162: What was the reasoning behind using 3 days?

As stated in the manuscript, during this time, calves were acclimating to their environment and the RFID tags began reporting locations at different times. Admittedly, the choice to use "3 days" specifically as an appropriate acclimation time period is a bit arbitrary, but it is a length of time we have used in previous work (Farthing et al. 2020).

R2.24) Contact and social network, and hazard modelling sections: Well described.

Thank you very much.

R2.25) Line 242: How often did that happen? Suggest providing specific numbers.

For susceptible individuals, it only happened once. One individual's colonization information was missing in week 1 of 2017. This is specified in Tables 2 & 3, as well as Figure 2a, where these data are described.

R2.26) Line 362: Suggest clarifying what "long periods of time" means. Can you be more specific?

Understood. The line now specifically refers to *E. coli*'s ability to survive months in soil without hosts.

R2.27) Line 364-365: An alternative explanation is that there were other factors that the authors did not account for that were influencing the results. Suggest providing examples of other factors not controlled for in this study that could have influenced these findings based on previous studies.

This is a good point. We now have a paragraph in the discussion, wherein we speculate on how including environmental covariates (e.g., precipitation, temperature, soil CFU load) could have improved our models. (*see* L407-L414).

R2.28) Line 366-367: Is there an incline in the pen? Otherwise I would not expect this to be the case. Suggest clarifying a bit more.

These lines specifically relate to our previous speculations on the biological relevance of observed environmental effects shown in our previous models. However, as noted above, after consideration of Reviewer 1's comments, the scope of our manuscript has been reduced and all mentions of environmental effects on colonization have been removed. Therefore, this information is now irrelevant.

R2.29) Lines 368-369: One of the references suggests that animals will favor concrete over soil. With such high-resolution movement data, the authors should be able to determine time spent on concrete vs. soil. Suggest clarifying this. Additionally, the authors should be able to determine the amount of movement vs. standing time, which could also help clarify this point. In this regard, suggest also comparing these differences between 2017 and 2018. Additionally, it was not totally clear to me whether individuals studied in 2017 were the same individuals studied in 2018? If different, this may also be influencing the findings and should be acknowledged.

These lines specifically relate to our previous speculations on the biological relevance of observed environmental effects shown in our models. However, as noted above, after consideration of Reviewer 1's comments, the scope of our manuscript has been reduced and all mentions of environmental effects on colonization have been removed. Therefore, this information is now irrelevant. That said, yes, our data resolution does allow us to examine where individuals were located and movement metrics.

2017 and 2018 study populations were not comprised of the same individuals. This has been made clearer in section 2.1.1 (i.e., the "Study population" subsection of the Methods), which now explicitly states that: "none of the 70 individuals in the 2017 study were retained for 2018 experiments."

R2.30) Lines 406-407: It was not clear from the introduction what these findings would be used for. Thus, listing this as a limitation is good but it reinforces the point that the authors need to clearly state who these findings are directed to. It seems a stretch to try and pitch this work to free-ranging animal systems.

As noted in our response to R2.7, testing or otherwise evaluating effects of intervention strategies was never within the scope of this study. While our work does present specific information about *E. coli* transmission dynamics in a cattle feedlot setting – and therefore may be used to inform transmission models related to this system – this manuscript is primarily intended to describe the methodological framework we used (i.e., combining transmission experiments with real-time location data to evaluate effects of direct contacts and social affiliation on transmission, and adapting Spiegel et al. (2016)'s methods to generate dyadic social networks from contact networks while controlling for environmental drivers of contact). We explicitly

state the primary take-aways from our work in the last paragraph of the Discussion section (*see* L460-L478).

R2.31) Lines 408-409: It would be good to provide more information here. Which seasons have more shedding? And does cattle behavior differ across seasons? If so, in which direction and what does this mean for your summer findings?

Understood. The lines now read: “Cattle are known to show increased *E. coli* shedding and carriage rates in summer months (Hancock et al. 1997; Cobbold et al. 2004; Ogden et al. 2004; Dawson et al. 2018), and may alter their behaviors (e.g., reduced activity and feed consumption, increased water consumption, etc.) in such a way that may change contact or social network structures in hot weather as well (Ray & Roubicek 1971; Lovarelli et al. 2020).”

Regarding cattle behavior, we explored the interplay between seasonal cattle behavior and *E. coli* colonization rates in depth in Dawson et al. (2018), which we cite in this paper.

R2.32) Lines 437-438: But the authors don't say how their findings help guide controlling transmission.

This sentence has been removed from the paper to reduce confusion.

R2.33) Figures and tables: Clear and informative.

Thank you very much.

Literature Cited

1. Cobbold R.N., D.H. Rice, M. Szymanski, D.R. Call, & D.D. Hancock. (2004). Comparison of shiga-toxigenic *Escherichia coli* prevalences among dairy, feedlot, and cow-calf herds in Washington State. *Applied Environmental Microbiology* 70(7):4375-4378. <https://dx.doi.org/10.1128%2FAEM.70.7.4375-4378.2004>.
2. Dawson, D.E., T.S. Farthing, M.W. Sanderson, & C. Lanzas. (2019). Transmission on empirical dynamic contact networks is influenced by data processing decisions. *Epidemics* 26:32-42. <https://doi.org/10.1016/j.epidem.2018.08.003>.
3. Dawson, D.E., J.H. Keung, M.G. Napoles, M.R. Vella, S. Chen, M.W. Sanderson, & C. Lanzas. (2018). Investigating behavioral drivers of seasonal shiga-toxigenic *Escherichia coli* (STEC) patterns in grazing cattle using an agent-based model. *PLoS One* 13(10): e0205418. <https://doi.org/10.1371/journal.pone.0205418>.
4. Diekmann, O., & J.A.P. Heesterbeek. (2000). *Mathematical Epidemiology of Infectious Diseases: Model Building, Analysis and Interpretation*. pp.54-55. Wiley, United Kingdom.
5. Hancock D.D., T.E. Besser, D.H. Rice, D.E. Herriott, & P.I. Tarr. (1997). A longitudinal

study of *Escherichia coli* O157 in fourteen cattle herds. *Epidemiology & Infection* 118(2):193-195. <https://doi.org/10.1017/S0950268896007212>.

6. Kleinlützum, D., G. Weaver, & D. Schley. (2013). Within-group contact of cattle in dairy barns and the implications for disease transmission. *Research in Veterinary Science* 95(2):425-429. <https://doi.org/10.1016/j.rvsc.2013.06.006>.
7. Lovarelli, D., A. Finzi, G. Mattachini, & E. Riva. (2020). A survey of dairy cattle behavior in different barns in northern Italy. *Animals* 10(4): 713. <https://doi.org/10.3390/ani10040713>.
8. Microsoft Corporation. (2016). Microsoft Office. Microsoft Corporation, Redmond, Washington, USA. <https://www.microsoft.com/en-us/microsoft-365/previous-versions/microsoft-office-2016>. Last accessed 07/09/2021.
9. Ray, D.E., & C.B. Roubicek. (1971). Behavior of feedlot cattle during two seasons. *Journal of Animal Science* 33(1): 72-76. <https://doi.org/10.2527/jas1971.33172x>.
10. Rocha, L.E.C., O. Terenius, I. Veissier, B. Meunier, & P.P. Nielsen. (2020). Persistence of sociality in group dynamics of dairy cattle. *Applied Animal Behaviour Science* 223(February 2020): 104921. <https://doi-org.prox.lib.ncsu.edu/10.1016/j.applanim.2019.104921>.
11. Spiegel, O., S.T. Leu, A. Sih, & C.M. Bull. (2016). Socially interacting or indifferent neighbors? Randomization of movement paths to tease apart social preference and spatial constraints. *Methods in Ecology and Evolution* 7(8): 971-979. <https://doi.org/10.1111/2041-210X.12553>.
12. Swain, D.L., & Bishop-Hurley G.J. (2007). Using contact logging devices to explore animal affiliations: quantifying cow-calf interactions. *Applied Animal Behaviour Science* 102(2007):1-11. <https://doi.org/10.1016/j.applanim.2006.03.008>.

Appendix B

Dear Editors,

Please consider our revised manuscript, “Combining epidemiological and ecological methods to quantify social effects on *E. coli* transmission,” for publication in *Royal Society Open Science*.

Once again, we sincerely appreciated the reviewer comments we received, and have incorporated them into our manuscript. In this draft we have updated our methods to include a bootstrap resampling procedure which allows us to better demonstrate the fit of our statistical models. Per the suggestion of Reviewer 1, we are also submitting additional supplemental materials. Finally, to ensure we are adhering to RSOS data guidelines, we deposited data sets in a Dryad repository.

Ultimately, our findings remain unchanged from the previously-submitted draft. That is to say, we continue to demonstrate that transmission experiments can be combined with real-time location data collection and processing procedures to create an effective framework for quantifying sociobehavioral effects on microparasite transmission. We also demonstrate how ecological methods for identifying sociality in animal populations can be adapted to characterize specific social relationships between individuals. Finally, we show that our sociobehavioral covariates were generally poor predictors of *E. coli* colonization in our study populations, but they can have significant ($p \leq 0.05$) effects on colonization rates. Significant effects on transmission however, were not consistent between similar populations, which highlights the potential variability inherent in these microparasite-host systems.

Due to the wide-ranging applicability of our methods and our contributions to the growing body of literature on sociobehavioral drivers of microparasite transmission, we continue to feel that our work is well suited for publication in *Royal Society Open Science*. We include a point-by-point response to reviewer comments below. Reviewer comments are presented in black, and our responses are in blue.

Thank you again for considering our revised manuscript.

Sincerely,

Trevor Farthing

Trevor Farthing, Ph.D.
Postdoctoral Research Scholar
North Carolina State University
Raleigh, NC 27695
tsfarthi@ncsu.edu

R1) Reviewer 1 Comments

R1.1) The authors removal of environmental co-variates addresses my main concerns with the previous submission. However, the new presentation of predictive performance of the new best-fit models raises further questions. This appears that it may be due to a minor mistake, but further review of this work depends on clarification of this key point.

Specifically, I am concerned that in Figure 2 (panels b,c) the authors have made the wrong comparison between the empirical and predicted survival probabilities. The panel claims to compare the weekly survival probability, but the trend and shape of the model predictions look more like cumulative survival probabilities (as plotted in panel a). Indeed, agreement between the model predictions and the empirical cumulative survival probability is much better and in line with the narrative in the discussion - with qualitatively good agreement for 2018 and poorer agreement for 2017 (driven by the exceptionally high observed transmission at the first time point).

If the current version of Figure 2 is correct the predictive performance of the fitted models is not even qualitatively correct. Such a systematic lack of fit would make interpretation of the estimated effects from the fitted models highly problematic - statistical significance is meaningless if the model cannot at least approximate the data. As presented this discrepancy is not "relatively poor" (Line 488), but completely unrepresentative.

Thank you once again for a detailed and thorough review. After reading your comments, we decided that the arrangement of data in Figures 2b & 2c were not as effective for communicating information about model fits as we had hoped. We have amended our methods and these figures to better describe the ability of our survival models to predict empirical trends in *E. coli* colonization. We now utilize a bootstrap resampling procedure to estimate population-level cumulative survival probabilities at each weekly timestep 100,000 times (see L338 – L344), and plot the mean predicted cumulative survival rates with a 95% bootstrap percentile confidence interval in the new iteration of Figures 2b & 2c. It appears as though the 2017 model overestimates baseline dyadic hazard rates. In contrast, the opposite is true for the high-weight 2018 candidates. These consistently underpredict baseline dyadic hazard rates. So when there are relative few infectious individuals, our model predicts an unlikely number of calves will avoid colonization. However, as shown in Figure 2c, our 2018 models can predict empirical trends in the first several weeks of the study period.

R1.2) Assuming the predictive checks have been plotted incorrectly I would still disagree with the authors conclusion that "Our results suggest that the number of close-proximity contacts and social relationships with shedding individuals are poor predictors for *E. coli* colonization in susceptible feedlot cattle overall, but can influence colonization rates." The former is clearly supported, but I would argue that the latter can at best not be ruled out rather than supported. For both years the additional information from the networks does not increase the predictive accuracy compared to the null model. The difference in AIC are small, which again I would argue indicates that we cannot rule out that the network structure impacts transmission risk there is no evidence to support it in this data. (i.e. the effect may be true but too small to be estimated in this study).

Actually, 2018 candidates with sociobehavioral predictors did outperform the null model. The difference in AIC values between the null model and the best-fitting 2018 candidate is 8.34. Furthermore, according to an AIC weight ratio, there is ≈ 65 times more empirical evidence to support the best-fitting candidate than the null model. In fact, the top 5 candidate models from the 2018 set have AIC weight ratios indicating 19–65 times more empirical support than the null model (*see* Table 4). We feel that this is sufficient evidence to make the statement that close-proximity contacts and social relationships can influence *E. coli* colonization rates in this system.

R1.3) This "null" result is absolutely still important on it's own given that our baseline assumption for infectious disease transmission is that social and physical contacts will increase the risk of transmission. The difficulty we have in confirming this intuitive assumption is for me a key open research question. For this system where transmission is mediated through the environment, the "null" result makes sense and I agree that using the environmental data could help to improve predictive ability. While I appreciate that developing new statistical methodology is out of the scope of this paper, I am a little disappointed that the environmental data was dropped completely. A null analysis comparing the predictive power of a model only trained using environmental co-variates should be straightforward and very much justified by the lack of importance of network structure suggested by these results.

We understand and are similarly disappointed that we needed to remove environmental covariates from the paper. Nevertheless, we feel it was the right decision given the comments we received on first draft of our manuscript.

R1.4) Response to R1.11. I would respectively disagree that the difference in sparsity between the two years is outside of the scope of this paper. If you wish to compare between the two years, then I would consider this an important (albeit supplemental) sensitivity analysis at least to show that downsampling to the same level in both datasets has no impact on results. (Given the lack of importance of the networks I would absolutely expect this to be the case, and absolutely would not consider a systematic exploration to be necessary).

Two plots have been added to Appendix S1 which detail effects of data sparsity on mean dyadic contact rates and the number of social relationships observed each week (*see* Figures S1-3 & S1-4).

R1.5) Line 240: The benefit of sparsity in location reporting during the day is presumably a smaller data set to work with? The latter motivation - i.e. that animals will be more active socially at these times is presumably the more important factor. When reading this I at first thought measurements at nighttime were more sparse, hence a reason to discount them. A little extra context (or simply focusing on the argument to focus on socially active hours) might make this clearer.

Good catch. The mention of data sparsity was unrelated to our decision to exclude nighttime hours in our analysis and was an unintentional holdover from a previous draft. The sentence has been edited to read: "Due to differences in the frequency of social behaviors associated with increased dyadic contact rates (e.g., allogrooming, headbutting, etc.) during daytime hours relative to nighttime hours – when animals would primarily be immobile and resting, we decided

to examine social relationships between calves during daytime hours only, when animals were most active.”

R1.6) Unless the github repository is to be archived/make read only after publication would be better to provide this in supplemental information (or simply rephrase to say that code can be used to recreate the full table).

As suggested, we now include these tables in supplementary materials (*see* Appendix S2).

R2) Reviewer 2 Comments

R2.1) Thank you for carefully and thoroughly addressing the issues raised in the previous review. No additional comments.

We're glad we were able to address all your comments to your satisfaction.